# Communication-Efficient Federated Group Distributionally Robust Optimization

**Zhishuai Guo, Tianbao Yang**\*
Department of Computer Science and Engineering
Texas A&M University
zhishguo@tamu.edu,tianbao-yang@tamu.edu

## Abstract

Federated learning faces challenges due to the heterogeneity in data volumes and distributions at different clients, which can compromise model generalization ability to various distributions. Existing approaches to address this issue based on group distributionally robust optimization (GDRO) often lead to high communication and sample complexity. To this end, this work introduces algorithms tailored for communication-efficient Federated Group Distributionally Robust Optimization (FGDRO). Our contributions are threefold: Firstly, we introduce the FGDRO-CVaR algorithm, which optimizes the average top-K losses while reducing communication complexity to $O(1/\epsilon^4)$, where $\epsilon$ denotes the desired precision level. Secondly, our FGDRO-KL algorithm is crafted to optimize KL regularized FGDRO, cutting communication complexity to $O(1/\epsilon^3)$. Lastly, we propose FGDRO-KL-Adam to to utilize Adam-type local updates in FGDRO-KL, which not only maintains a communication cost of $O(1/\epsilon^3)$ but also shows potential to surpass SGD-type local steps in practical applications. The effectiveness of our algorithms has been demonstrated on a variety of real-world tasks, including natural language processing and computer vision.

## 1 Introduction

Federated learning enables effective model training without the need to share raw data [38, 48]. It is essential in contexts where data privacy and ownership are paramount, such as in inter-hospital collaborations [54] and mobile device networks [20]. However, clients often have data of varying volumes and distinct distributions, which poses notable challenges in maintaining generalization behavior [50, 27]. Generalization here refers to the model's ability to perform consistently across different clients, including those that have not participated in the training [23, 79].

In this study, we tackle the issue using federated group distributionally robust optimization (FGDRO), formulated as follows:

$$\min_{\mathbf{w}} F(\mathbf{w}) := \max_{\mathbf{p} \in \Delta_N} \sum_{i=1}^{N} \mathbf{p}_i \ell_i(\mathbf{w}) - \lambda \phi(\mathbf{p}). \tag{1}$$

Here, $\mathbf{w}$ denotes a machine learning model, and $N$ represents the number of clients. For each client $i$, $\mathcal{D}_i$ represents its local data distribution, and $\ell_i(\mathbf{w}) = \mathbb{E}_{\mathbf{z} \sim \mathcal{D}_i} \ell(\mathbf{w}; \mathbf{z})$ represents the loss calculated from that local distribution. $\Delta_N$ denotes a $N$-dimensional simplex, which constrains $\sum_i p_i = 1$. The vector $\mathbf{p} = [\mathbf{p}_1, ..., \mathbf{p}_N]$ comprises the weights assigned to each of the $N$ clients. The function $\phi(\mathbf{p})$ acts as a regularization term, with $\lambda > 0$ being an adjustable parameter. This framework aims to assign higher weights to machines with greater losses while discouraging substantial deviations of these weights from a specified distribution.

---

\*Corresponding Author

38th Conference on Neural Information Processing Systems (NeurIPS 2024).

Our study concentrates on two particular forms of the regularization term $\phi$, which are well-established regularization techniques [11, 39], each suited to different tasks and data distributions. Specifically, CVaR is defined as $\phi(\mathbf{p}) = \mathbb{I}_{[0,1/K]}(\mathbf{p})$. In this scenario, $\phi(\mathbf{p})$ is set to 0 if each weight $\mathbf{p}_i$ falls within the range of $[0, 1/K]$, and is infinite otherwise. FGDRO-CVaR focuses on optimizing for worst-case scenarios or the average of the worst-case losses, making it particularly effective in high-stakes applications like healthcare and finance, where avoiding extreme losses is crucial. However, it can be sensitive to outliers or malicious client attacks. FGDRO-KL, on the other hand, uses Kullback-Leibler (KL) divergence, expressed as $\phi(\mathbf{p}) = \sum_{i=1}^{N} \mathbf{p}_i \log(N\mathbf{p}_i)$. This version of $\phi$ penalizes deviations of the weight distribution $\mathbf{p}$ from a uniform distribution. Fundamentally, when $\phi(\mathbf{p})$ is strongly convex, as in the case of KL divergence, $F(\mathbf{w})$ can enjoy a smoothness property, while non-strongly convex $\phi(\mathbf{p})$ would result in non-smooth $F(\mathbf{w})$ [6]. In contrast to CVaR, KL is a softer regularizer to promote smoother and more stable learning. Thus, it can be beneficial in scenarios where robustness to outliers or malicious clients is needed. FGDRO-KL-Adam further enhances FGDRO-KL by incorporating Adam-type updates.

Table 1: Comparison of communication cost and sample complexity on each machine to achieve $\epsilon$-stationary point or near to $\epsilon$-stationary point, where $\epsilon$-stationary point has a (sub-)gradient $\|\partial F(\mathbf{w})\|^2 \leq \epsilon^2$. NDP-SONT denotes the naive deployment of the SONT algorithm [22] in a federated environment, communicating in all iterations.

| | FGDRO with a CVaR constraint | | FGDRO with a KL regularization | |
|---|---|---|---|---|
| | Communication Complexity | Sample Complexity | Communication Complexity | Sample Complexity |
| DRFA [11] | $O\left(\frac{1}{\epsilon^{12}}\right)$ | $O\left(\frac{1}{\epsilon^{16}}\right)$ | $O\left(\frac{1}{\epsilon^{12}}\right)$ | $O\left(\frac{1}{\epsilon^{16}}\right)$ |
| DR-DSGD [30] | $-$ | $-$ | $O(\frac{1}{\epsilon^3})$ | $O(\frac{1}{\epsilon^6})$ |
| NDP-SONT [22] | $O\left(\frac{1}{\epsilon^6}\right)$ | $O\left(\frac{1}{\epsilon^6}\right)$ | $O\left(\frac{1}{\epsilon^6}\right)$ | $O\left(\frac{1}{\epsilon^6}\right)$ |
| This Work | $O\left(\frac{1}{\epsilon^4}\right)$ | $O\left(\frac{1}{\epsilon^8}\right)$ | $O\left(\frac{1}{\epsilon^3}\right)$ | $O\left(\frac{1}{\epsilon^4}\right)$ |

Previous research addressing these optimization problems in federated learning has struggled with high communication and sample complexity issues, which are basically due to inefficient updates of $\mathbf{p}$ on local machines. For example, Deng et al. [11] examined a particular case of the general formula (1) using a CVaR constraint with $K = 1$, and developed an algorithm for KL regularization as well. They update $\mathbf{p}$ only in global communication rounds, while local steps only optimize the local loss function using stochastic gradient descent (SGD). To achieve a $\epsilon$-stationary point or a point near to an $\epsilon$-stationary point, where an $\epsilon$-stationary point has a (sub)gradient $\|\partial F(\mathbf{w})\|^2 \leq \epsilon^2$, their methods required a communication cost of $O(1/\epsilon^{12})$ and a sample complexity of $O(1/\epsilon^{16})$ on each client. For FGDRO with KL regularization, [30] achieves a communication cost of $O(1/\epsilon^3)$, but it requires the use of large data batches in local update steps in order to get good approximation for the surrogate of $\mathbf{p}$, resulting in a total sample complexity of $O(1/\epsilon^6)$ per machine.

To overcome these limitations, this paper presents specialized algorithms FGDRO-CVaR and FGDRO-KL for FGDRO with CVaR constraint and KL regularization, respectively. Instead of dealing with the constrained primal-dual formulation in (1), we consider their equivalent forms with a compositional structure that get rid of the high-dimensional constrained variable $\mathbf{p}$. We summarize the complexity results in Table 1.

**For FGDRO with CVaR constraint**, we are the first to consider a constraint-free equivalent form and develop a communication-efficient algorithm for it, significantly reducing communication costs, as shown in Table 1. In addition to sharing machine learning models, we only introduce an additional scalar threshold to select participants in each round, minimizing additional costs. The equivalent compositional form is a non-smooth two-level compositional function with one auxiliary variable $s$, which works as a threshold. Only machines whose local losses are greater than $s$ are supposed to contribute to updating the model. In this way, we can simply update the constraint-free scalar variable $s$ locally in each client and average $s$ in communication rounds. However, we do face challenges with non-smooth compositional optimization problems. Our first algorithm FGDRO-CVaR effectively

addresses this issues and achieves a communication cost of $O(1/\epsilon^4)$ and a sample complexity of $O(1/\epsilon^8)$ on each machine.

**For FGDRO with KL regularization**, while previous literature has explored constraint-free compositional reformulations, they often require large batch sizes on each machine to estimate gradients, making this approach impractical and leading to high sample complexity. In contrast, we utilize moving averages that can work with smaller data batches while still providing accurate gradient estimates, enhancing the efficiency of our method. The equivalent compositional form we consider is a smooth three-level compositional function. In this case, the weights for the clients depend on both local loss functions and global loss functions. We use moving average estimators of these statistics and update the estimators locally. In communication rounds, in addition to averaging the model $\mathbf{w}$, the machines will average the estimator of the global loss function. We have reduced the communication cost and the computation cost compared to the literature as presented in Table 1.

To further enhance our approach, we have developed **an adaptive algorithm for solving FGDRO with KL regularization**, named FGDRO-KL-Adam. Stochastic adaptive methods apply variable step sizes for each coordinate based on historical gradient information, often yielding better results than non-adaptive techniques, as evidenced by a wealth of research [13, 36, 49, 69]. In federated learning, while Reddi et al. [59] have developed a federated adaptive algorithm and shown its effectiveness in various tasks. However, it limits adaptive steps to global updates on the server, with local updates relying on standard SGD, which may lead to suboptimal results. Moreover, their method is primarily designed for Empirical Risk Minimization (ERM) and is not applicable to address compositional optimization problems. Our FGDRO-KL-Adam allows local updates to use Adam-type updates, which introduces the challenge of handling unbiased gradients, further complicated by the use and updating of the second-order moment. To this end, we update the first-order momentum and second-order momentum locally and then average them globally during communication rounds. Moreover, our analysis carefully manages the moving estimates of the first and second-order moments, ensuring that the solution provably converges.

Our FGDRO-KL-Adam enables local updates with Adam-type methods, which raises the challenge of maintaining unbiased gradients, especially with the adjustment of the second-order moment. Our analysis meticulously handles the moving estimates of both first and second-order moments to guarantee provable convergence. The first-order momentum and second-order momentum are updated locally and then averaged during communication rounds.

In summary, our paper contributes in three main areas. First, our FGDRO-CVaR algorithm greatly reduces both communication costs and sample complexity for FGDRO with CvaR constraint problems. Second, our FGDRO-KL algorithm achieves a better sample complexity while maintaining the same communication costs as the existing results. Third, our FGDRO-KL-Adam integrates adaptive step sizes with Adam-type updates, which has the potential to surpass the performance of conventional SGD-based approaches. Extensive testing on diverse real-world datasets has shown that our approach achieves superior performance while substantially reducing communication overhead.

## 2  Related Work

Federated learning has gained significant attention due to its potential to train machine learning models using data from various sources while ensuring data privacy [38, 48, 54]. Two central challenges to this field are communication cost and client heterogeneity, which have been extensively explored in the literature [63, 76, 77, 73, 62, 34, 64, 2, 31, 68, 4, 33, 35, 71, 70, 34, 18]. This section will dive into the body of literature that focuses on these specific challenges.

**Non-IID Clients in Federated Learning (FL)**   One of the key challenges in Federated Learning (FL) is managing client heterogeneity, particularly the issue of non-IID (nonindependently and identically distributed) data across client networks. Efforts to overcome the negative implications of data diversity have led to the development of model personalization techniques [47, 10, 44, 82, 45, 43, 75, 19, 41]. However, these approaches face challenges when dealing with data from unseen or unidentifiable groups. For a comprehensive examination of the challenges and strategies concerning non-IID clients in federated learning, the readers are directed to [27].

**Federated Group Distributionally Robust Optimization**   Since Group Distributionally Robust Optimization has shown effectivenss in addressing non-iid data in centralized setting [14, 51, 12, 56],

previous research has investigated Federated Group Distributionally Robust Optimization (FGDRO) to address the challenges posed by non-IID clients in federated settings [50, 11]. The DRFA algorithm [11] focuses on a specific instance of (2), applying a CVaR constraint on $\mathbf{p}$ with $K = 1$. It samples machines based on updated probabilities to allow local updates, reducing the need for communication, with these probabilities managed by a central server. However, this approach results in significant communication costs of $O(1/\epsilon^{12})$ and sample complexity of $O(1/\epsilon^{16})$ per machine to achieve an $\epsilon$-stationary point. Recent developments in [22] introduced algorithms for handling Group DRO with a CVaR constraint in centralized settings, but adapting these to federated learning entails substantial communication overheads of $O(1/\epsilon^6)$. Moreover, [78] introduced the SCAFF-PD algorithm, which is only applicable in convex scenarios and requires the use of the complete dataset in each training round. Regarding KL regularization, [30] achieved a communication cost of $O(1/\epsilon^3)$, but required the use of large data batches, resulting in a total sample complexity of $O(1/\epsilon^6)$ per machine. [80] has studied FGDRO with KL regularization in a decentralized setting, which would incur a communication cost of $O(1/\epsilon^4)$ if directly applied to a centralized federated learning setting.

**Federated Adaptive Algorithm** Stochastic adaptive methods for minimization in non-convex stochastic optimization have garnered significant interest in recent years [13, 36, 49, 69, 42, 84, 65, 7, 46, 25, 16, 81]. These methods, known for assigning unique step sizes to each coordinate, often outperform their non-adaptive counterparts. In federated learning, Reddi et al. [59] have advanced the field with an adaptive algorithm. However, their methodology predominantly applies adaptive steps at the global server level, with local updates still dependent on SGD. This could lead to suboptimal performance. Furthermore, their approach was tailored for Empirical Risk Minimization (ERM) problems and could not be applied for problems considered in this work.

## 3 Preliminaries

A function $f$ is $C$-Lipschitz if $f(\mathbf{x}) - f(\mathbf{y}) \leq C\|\mathbf{x} - \mathbf{y}\|$. A differentiable function $f$ is $L$-smooth if $\|\nabla f(\mathbf{x}) - \nabla f(\mathbf{y})\| \leq L\|\mathbf{x} - \mathbf{y}\|$, where $\nabla f(\mathbf{x})$ denotes the gradient. For a non-differentiable function $f$, its subdifferential $\partial f(\mathbf{x})$ is defined as a set of all subgradients as $\partial f(\mathbf{x}) = \{\mathbf{v}|f(\mathbf{y}) \geq f(\mathbf{x}) + \langle\mathbf{v}, \mathbf{y} - \mathbf{x}\rangle + o(\|\mathbf{y} - \mathbf{x}\|)\}$ as $\mathbf{y} \to \mathbf{x}$. When the context is clear, we also overload the notation $\partial f(\mathbf{x})$ to denote one subgradient from the subdifferential set. We use $\nabla f(\mathbf{x}; \mathbf{z})$ or $\partial f(\mathbf{x}; \mathbf{z})$ to represent an unbiased estimator of gradient or subgradient with a randomly drawn sample $\mathbf{z}$. Additionally, a function $f$ is $\rho$-weakly convex if $f(\mathbf{x}) \geq f(\mathbf{y}) + \langle\partial f(\mathbf{y}), \mathbf{y} - \mathbf{x}\rangle - \frac{\rho}{2}\|\mathbf{y} - \mathbf{x}\|^2$.

For a smooth function $f(\mathbf{x})$, $\mathbf{x}$ is an $\epsilon$-stationary point if $\|\nabla f(\mathbf{x})\|^2 \leq \epsilon^2$. For non-smooth functions, $\mathbf{x}$ is an $\epsilon$-stationary point if $\|dist(\mathbf{0}, \partial f(\mathbf{x}))\|^2 \leq \epsilon^2$, where $dist(\mathbf{x}, S) = \min_{\mathbf{x}' \in S}\|\mathbf{x} - \mathbf{x}'\|_2$ measures the distance between a point $\mathbf{x}$ and a set $S$. For non-smooth functions, since it is usually difficult or even impossible to find an $\epsilon$-stationary point, we instead seek an $\epsilon$-near stationary point.

**Definition 3.1.** $\mathbf{x}$ is an $\epsilon$-near stationary point of $f(\cdot)$ if $\exists \mathbf{x}'$ such that $\|\mathbf{x} - \mathbf{x}'\|_2 \leq \epsilon$ and $dist(\mathbf{0}, \partial f(\mathbf{x}')) \leq \epsilon$.

## 4 FGDRO-CVaR

In this section, we present our algorithm designed to tackle Federated Group Distributionally Robust Optimization (FGDRO) with a CVaR constraint. This problem poses substantial challenges due to the complexity of both CVaR and simplex constraints. Typically, during local updates, individual machines do not have access to adequate information to appropriately adjust the weight vector $\mathbf{p}$. Prior approaches, such as the one proposed by [11], mitigate this issue by updating $\mathbf{p}$ during global communication rounds, but results in slower convergence rates. To this end, we reformulate the problem into an equivalent two-level compositional optimization problem without constraints:

$$\min_{\mathbf{w}} \min_{s} F(\mathbf{w}, s) := \frac{1}{N}\sum_{i=1}^{N} f(g_i(\mathbf{w}) - s) + \frac{K}{N}s. \tag{2}$$

where $f(\cdot) = (\cdot)_+$ and $g_i(\mathbf{w}) = \mathbb{E}_{\mathbf{z}\sim\mathcal{D}_i}\ell(\mathbf{w}, \mathbf{z})$ is a local loss and $s$ is intended to serve as a threshold value. With $s = \arg\min_{s'} \frac{1}{N}\sum_{i=1}^{N} f(g_i(\mathbf{w}) - s') + \frac{K}{N}s'$, only the $K$ clients with the highest losses will have losses greater than $s$ [53, 83]. During the training phase, $K$ clients with the highest losses are expected to predominantly influence the optimization process.

The formulation (2) replaces the constrained high-dimensional vector $\mathbf{p}$ with a single unconstrained scalar variable $s$. However, this adjustment introduces new challenges due to the compositional structure and the non-smooth nature of the outer function $f$. As a result, it is biased to estimate subgradient $\partial f(g_i(\mathbf{w}) - s)\nabla g_i(\mathbf{w})$ using a batch of samples. To address this, it is common practice to create an accurate estimate of $g_i(\mathbf{w})$ [67, 24, 22, 66, 32, 55, 57]. Specifically, we employ a moving average $\mathbf{u}$ as our accurate estimate:

$$u_{i,t}^r = (1 - \beta_1)u_{i,t-1}^r + \beta_1 \ell(\mathbf{w}; \mathbf{z}_{i,t}^r). \tag{3}$$

And then the estimators for sub-gradient of $\mathbf{w}$ and $s$, namely $\mathbf{m}$, $v$ are computed using $u$. It is notable that for $s$, it is updated locally using local data and averaged between clients in communication rounds. It will converge alongside $\mathbf{w}$ to an $\epsilon$-near stationaty point. This is a fundamental reason why our method achieves a lower communication complexity compared to [11], as the latter can only update the weight variables at the server node in the communication rounds.

---

**Algorithm 1** FGDRO-CVaR

---

1: Initialization: $\bar{\mathbf{w}}^1$, $\bar{s}^1 = 0$, $u_{i,t}^0 = 0$,
2: **for** $r = 1, ..., R$ **do**
3:     $\mathbf{w}_{i,0}^r = \bar{\mathbf{w}}^r$, $s_{i,0}^r = \bar{s}^r$, $u_{i,0}^r = u_{0,I}^{r-1}$
4:     **for** $t = 1, ..., I$ **do**
5:         Each machine samples data $\mathbf{z}_{i,t}^r$
6:         $u_{i,t}^r = (1 - \beta_1)u_{i,t-1}^r + \beta_1 \ell(\mathbf{w}_{i,t-1}^r; \mathbf{z}_{i,t}^r)$
7:         $v_{i,t}^r = -\partial f(u_{i,t}^r - s_{i,t-1}^r) + \frac{K}{N}$, and $s_{i,t}^r = s_{i,t-1}^r - \eta_2 v_{i,t}^r$
8:         $\mathbf{m}_{i,t}^r = \partial f(u_{i,t}^r - s_{i,t-1}^r)\nabla \ell(\mathbf{w}_{i,t-1}^r, \mathbf{z}_{i,t}^r)$, and $\mathbf{w}_{i,t}^r = \mathbf{w}_{i,t-1}^r - \eta_1 \mathbf{m}_{i,t}^r$
9:     **end for**
10:     $\bar{\mathbf{w}}^{r+1} = \frac{1}{N}\sum_{i=1}^N \mathbf{w}_{i,I}^r$, $\bar{s}^{r+1} = \frac{1}{N}\sum_{i=1}^N s_{i,I}^r$
11: **end for**
12: Output: $\tilde{\mathbf{w}} = \frac{1}{N}\sum_{i=1}^N \mathbf{w}_{i,t'}^{r'}$, where $r'$ and $t'$ are sampled from $[1, R]$ and $[1, I]$, respectively.

---

We present the formalization of our FGDRO-CVaR method in Algorithm 1. Next, we show the convergence results of FGDRO-CVaR . We make the following assumptions regarding problem (2).

**Assumption 4.1.** (1) $\forall i$ and $\forall \mathbf{z} \in D_i$, $\ell(\cdot, \mathbf{z})$ is $C_g$-Lipschitz and $L_g$-smooth. (2) $\mathbb{E}_{\mathbf{z} \in \mathcal{D}_i} \|\nabla \ell(\mathbf{w}; \mathbf{z}) - \nabla g_i(\mathbf{w})\|^2 \leq \sigma^2$, $\mathbb{E}_{\mathbf{z} \in \mathcal{D}_i} \|\ell(\mathbf{w}; \mathbf{z}) - g_i(\mathbf{w})\|^2 \leq \sigma^2$.

**Remark.** The first assumption about Lipschitz continuity and smoothness of $g_i$ is standard in compositional optimization [22, 66, 67]. The second assumption of bounded variance is also common. Assumption 4.1 leads to $F(\mathbf{w}, s)$ being weakly convex, which is a key step in the analysis as shown in Appendix A.2

The behavior of the estimator $u$ is examined through the following lemma.

**Lemma 4.2.** *Under Assumption 4.1, by setting* $\eta = O\left(1/(R)^{3/2}\right)$, $\beta_1 = O\left(1/R\right)$, $I = O(R)$, *Algorithm 1 ensures that*

$$\mathbb{E}\|u_{i,t}^r - g_i(\bar{\mathbf{w}}_t^r)\|^2 \leq (1 - \beta_1)\mathbb{E}\|u_{i,t-1}^r - g_i(\bar{\mathbf{w}}_{t-1}^r)\|^2 + 2\beta_1^2\sigma^2 + 3\beta_2\eta^2 I^2 C_g^2 + \frac{5}{\beta_1}C_g^2\|\bar{\mathbf{w}}_t^r - \bar{\mathbf{w}}_{t-1}^r\|^2.$$

The convergence result of FGDRO-CVaR is given in the following theorem.

**Theorem 4.3.** *Under Assumption 4.1, by setting* $\eta = O\left(1/(R)^{3/2}\right)$, $\beta_1 = O\left(1/R\right)$, $I = O(R)$ *and* $\hat{\rho} = 2L_g$, *the Algorithm 1 ensures that for the output* $(\tilde{\mathbf{w}}, \tilde{s})$, *there exists* $(\mathbf{w}', s')$ *that*

$$\|dist(\mathbf{0}, F(\mathbf{w}', s'))\|^2 \leq 1/\hat{\rho}(\|\tilde{\mathbf{w}} - \mathbf{w}'\|_2^2 + \|\tilde{s} - s'\|_2^2) \leq O\left(\frac{1}{R^{1/2}}\right). \tag{4}$$

**Remark.** The analysis has utilized Moreau envelop to address the nonsmooth issue [52, 9]. To achieve an $\epsilon$-near stationary point of $F(\cdot)$, we need to set $R = O(1/\epsilon^4)$ and $I = O(1/\epsilon^4)$, and thus the sample complexity on each machine is $RI = O(1/\epsilon^8)$. The total sample complexity of $O(n/\epsilon^6)$

by [22] is achieved by the STORM estimator [8] which incurs additional memory and computational costs due to the requirement of computing gradients using two models at each iteration. Without the STORM estimator, [22] would exhibit a total sample complexity of $O(n/\epsilon^8)$. When deployed in a federated setting, the complexity for each machine would be $O(1/\epsilon^8)$, aligning with our results and demonstrating that our approach achieves a linear speed-up in terms of number of machines. Additionally, although we aggregate certain scalar variables (in FGDRO-CVaR, the scalar variable s; in FGDRO-KL and FGDRO-KL-Adam to be presented later, the scalar variable v), similar to the technique in the Remark 3.1 of [61], we can aggregate these variables using homomorphic encryption, ensuring that their exact values remain confidential.

# 5 FGDRO-KL

In this section, we present our FGDRO-KL algorithm for solving problem (1) with a KL regularization. Unlike the CVaR constraints that focus on the top K clients, KL regularization takes into account all clients, assigning them varying weights. Additionally, FGDRO with KL regularization is smooth, and strongly concave with respect to $\mathbf{p}$. Nevertheless, it is subject to the simplex constraint on $\mathbf{p}$. To address this, we use an equivalent form derived from the KKT conditions, as referenced in [56, 30]:

$$\min_{\mathbf{w}} F(\mathbf{w}) = \lambda \log(\frac{1}{N} \sum_{i=1}^{N} \exp(\mathbb{E}_{\mathbf{z} \sim \mathcal{D}_i} \ell(\mathbf{w}; \mathbf{z})/\lambda)). \tag{5}$$

This formulation eliminates the constrained vector $\mathbf{p}$, and $F(\mathbf{w})$ is smooth since KL regularization is strongly concave [6]. However, this formulation has a three-level composition structure and thus, using a batch of data in a three-level composition can result in biased gradient estimation. Furthermore, the gradients on one machine are depend on other machines.

Specifically, we denote $g_i(\mathbf{w}) = \exp(\mathbb{E}_{\mathbf{z} \in \mathcal{D}_i} \ell(\mathbf{w}; \mathbf{z})/\lambda)$ and $g(\mathbf{w}) = \frac{1}{N} \sum_{i=1}^{N} g_i(\mathbf{w})$, with $\ell(\mathbf{w}; \mathcal{D}_i) = \mathbb{E}_{\mathbf{z} \in \mathcal{D}_i} \ell(\mathbf{w}; \mathbf{z})$, then, the gradient of $F(\mathbf{w})$ in (5) is given by:

$$\nabla F(\mathbf{w}) = \frac{1}{N} \sum_{i=1}^{N} \frac{g_i(\mathbf{w})}{g(\mathbf{w})} \nabla \ell(\mathbf{w}; D_i). \tag{6}$$

It is crucial to recognize that the gradient for machine $i$, i.e., $\nabla \ell(\mathbf{w}; \mathcal{D}_i)$, is scaled by $g_i(\mathbf{w})/g(\mathbf{w})$. This scaling indicates that machines experiencing larger loss functions exert more influence over the training process. To mitigate the biased gradient estimation, we approximate $g_i(\mathbf{w})$ and $g(\mathbf{w})$ based on moving average estimators $u$ and $v$. On each machine, $u$ serves as a moving average estimator for the local loss function $\ell(\mathbf{w}; \mathcal{D}_i)$, with $\exp(u_{i,t}^r/\lambda)$ providing a local approximation of $g_i(\mathbf{w})$. $u$ is updated and maintained locally without need for averaging during communication rounds. $v$ estimates the global statistic $g(\mathbf{w})$, and is updated locally but averaged during global communication rounds. Subsequently, a moving average estimator of the gradient, denoted as $\mathbf{m}$ is constructed using $u$ and $v$. For specific update rules, please refer to Algorithm 2.

For analysis, we make the following assumptions regarding problem (1) with a KL regularization:

**Assumption 5.1.** (1) $\forall i$ and $\forall \mathbf{z} \in D_i$, $g_i(\cdot)$ is $C_g$-Lipschitz and $L_g$-smooth. (2) $\mathbb{E}_{\mathbf{z} \in \mathcal{D}_i} \|\nabla \ell(\mathbf{w}; \mathbf{z}) - \nabla \ell(\mathbf{w}; \mathcal{D}_i)\|^2 \leq \sigma^2$, $\mathbb{E}_{\mathbf{z} \in \mathcal{D}_i} \|\ell(\mathbf{w}; \mathbf{z}) - \ell(\mathbf{w}; \mathcal{D}_i)\|^2 \leq \sigma^2$. (3) $f$ is $C_f$-Lipschitz and $L_f$-smooth. (4) $\forall i$ and $\forall \mathbf{z} \in D_i$, $0 \leq \ell(\cdot; \mathbf{z}) \leq C_0$, $\ell(\cdot)$ is $C_\ell$-Lipschitz and $L_\ell$-smooth.

The behavior of the $u$ and $v$ estimators can be bounded similar to the previous section and are shown in Appendix B. The estimator $\mathbf{m}$ for gradient can be bounded as

**Lemma 5.2.** Under Assumption 5.1, with proper constants $C_1$ and $G$, by setting $\eta = O\left(\frac{1}{\sqrt{RI}}\right)$, $\beta_1 = O\left(\frac{1}{\sqrt{RI}}\right)$, the Algorithm 2 ensures that

$$\|\bar{\mathbf{m}}_t^r - \nabla F(\bar{\mathbf{w}}_t^r)\|^2 \leq (1 - \frac{\beta_3}{2})\|\bar{\mathbf{m}}_{t-1}^r - \nabla F(\bar{\mathbf{w}}_{t-1}^r)\|^2 + \beta_3 C_\ell^2 C_1^2 \frac{1}{N} \sum_{i=1}^{N} \|u_{i,t-1}^r - \ell(\bar{\mathbf{w}}_{t-1}^r; \mathcal{D}_i)\|^2$$

$$+ \beta_3 C \|\bar{v}_t^r - g(\bar{\mathbf{w}}_t^r)\|^2 + 3\eta \|\nabla F(\bar{\mathbf{w}}_{t-1}^r)\|^2 + \beta_3^2 C_1^2 \frac{\sigma^2}{N} + 2\beta_3 C_1^2 L_\ell^2 \eta^2 I^2 G^2.$$

---

**Algorithm 2** FGDRO-KL

---

1: Initialization: $\bar{\mathbf{w}}^1, u_{i,I}^0, \bar{v}^1, \bar{\mathbf{m}}^1$
2: **for** $r = 1, ..., R$ **do**
3:    $\mathbf{w}_{i,0}^r = \bar{\mathbf{w}}^r, \mathbf{m}_{i,0}^r = \bar{\mathbf{m}}^r, u_{i,0}^r = u_{i,I}^{r-1}, v_{i,0}^r = \bar{v}^r$
4:    **for** $t = 1, ..., I$ **do**
5:       Each machine samples data $\mathbf{z}_{i,t}^r$
6:       $u_{i,t}^r = (1 - \beta_1)u_{i,t-1}^r + \beta_1 \ell(\mathbf{w}_{i,t-1}^r; \mathbf{z}_{i,t}^r)$, and $v_{i,t}^r = (1 - \beta_2)v_{i,t-1}^r + \beta_2 \exp(u_{i,t}^r/\lambda)$
7:       $\mathbf{h}_{i,t}^r = \frac{\exp(u_{i,t}^r)}{v_{i,t}^r} \nabla \ell(\mathbf{w}_{i,t-1}^r; \mathbf{z}_{i,t}^r)$, and $\mathbf{m}_{i,t}^r = (1 - \beta_3)\mathbf{m}_{i,t-1}^r + \beta_3 \mathbf{h}_{i,t}^r$
8:       $\mathbf{w}_{i,t}^r = \mathbf{w}_{i,t-1}^r - \eta \mathbf{m}_{i,t}^r$
9:    **end for**
10:   $\bar{\mathbf{w}}^{r+1} = \frac{1}{N}\sum_{i=1}^{N}\mathbf{w}_{i,I}^r, \bar{v}^{r+1} = \frac{1}{N}\sum_{i=1}^{N}v_{i,I}^r$, and $\bar{\mathbf{m}}^{r+1} = \frac{1}{N}\sum_{i=1}^{N}\mathbf{m}_{i,I}^r$
11: **end for**
12: Output: $\tilde{\mathbf{w}} = \frac{1}{N}\sum_{i=1}^{N}\mathbf{w}_{i,t'}^{r'}$, where $r'$ and $t'$ are sampled from $[1, R]$ and $[1, I]$, respectively.

---

---

**Algorithm 3** FGDRO-KL-Adam

---

1: Initialization: $\bar{\mathbf{w}}^1, u_{i,I}^0, \bar{v}^1, \bar{\mathbf{m}}^1, \bar{\mathbf{q}}^1$
2: **for** $r = 1, ..., R$ **do**
3:    $\mathbf{w}_{i,0}^r = \bar{\mathbf{w}}^r, \mathbf{m}_{i,0}^r = \bar{\mathbf{m}}^r, \mathbf{q}_{i,0}^r = \bar{\mathbf{q}}^r, u_{i,0}^r = u_{i,I}^{r-1}$, and $v_{i,0}^r = \bar{v}^r$
4:    **for** $t = 1, ..., I$ **do**
5:       Each machine samples data $\mathbf{z}_{i,t}^r$
6:       $u_{i,t}^r = (1 - \beta_1)u_{i,t-1}^r + \beta_1 \ell(\mathbf{w}_{i,t-1}^r; \mathbf{z}_{i,t}^r)$, and $v_{i,t}^r = (1 - \beta_2)v_{i,t-1}^r + \beta_2 \exp(u_{i,t}^r/\lambda)$
7:       $\mathbf{h}_{i,t}^r = \frac{\exp(u_{i,t}^r)}{v_{i,t}^r} \nabla \ell(\mathbf{w}_{i,t-1}^r; \mathbf{z}_{i,t}^r)$
8:       $\mathbf{m}_{i,t}^r = (1 - \beta_3)\mathbf{m}_{i,t-1}^r + \beta_3 \mathbf{h}_{i,t}^r$, and $\mathbf{q}_{i,t}^r = (1 - \beta_4)\mathbf{q}_{i,t-1}^r + \beta_4 (\mathbf{h}_{i,t}^r)^2$
9:       $\mathbf{w}_{i,t}^r = \mathbf{w}_{i,t-1}^r - \eta \frac{\mathbf{m}_{i,t}^r}{\sqrt{\mathbf{q}_{i,t}^r + \tau}}$
10:   **end for**
11:   $\bar{\mathbf{w}}^{r+1} = \frac{1}{N}\sum_{i=1}^{N}\mathbf{w}_{i,I}^r, \bar{v}^{r+1} = \frac{1}{N}\sum_{i=1}^{N}v_{i,I}^r, \bar{\mathbf{m}}^{r+1} = \frac{1}{N}\sum_{i=1}^{N}\mathbf{m}_{i,I}^r$ and $\mathbf{q}^{r+1} = \frac{1}{N}\sum_{i=1}^{N}\mathbf{q}_{i,I}^r$
12: **end for**
13: Output: $\tilde{\mathbf{w}} = \frac{1}{N}\sum_{i=1}^{N}\mathbf{w}_{i,t'}^{r'}$, where $r'$ and $t'$ are sampled from $[1, R]$ and $[1, I]$, respectively.

---

The precisions of the $u, v, \mathbf{m}$ estimators depend on each other. The idea is to get $\mathbb{E}\|u_{i,t}^r - \ell(\bar{\mathbf{w}}_t^r; \mathcal{D}_i)\|^2$, $\mathbb{E}\|\bar{v}_t^r - g(\bar{\mathbf{w}}_t^r)\|^2, \|\bar{\mathbf{m}}_t^r - \nabla F(\bar{\mathbf{w}}_t^r)\|^2$ and $\mathbb{E}\|\nabla F(\tilde{\mathbf{w}})\|^2$ jointly converge, and then we finally have the following theorem to guarantee the convergence:

**Theorem 5.3.** *Under Assumption 5.1, by setting* $\eta = O\left(\frac{1}{\sqrt{RI}}\right)$, $\beta_1 = O\left(\frac{1}{\sqrt{RI}}\right)$, $I = R^{1/3}$, *Algorithm 2 ensures that*

$$\mathbb{E}\|\nabla F(\tilde{\mathbf{w}})\|^2 \leq O\left(\frac{1}{R^{2/3}}\right). \tag{7}$$

**Remark.** To achieve an $\epsilon$-stationary point, i.e., $\|\nabla F(\tilde{\mathbf{w}})\|^2 \leq \epsilon^2$, we need to set $R = O(1/\epsilon^3)$, $I = O(1/\epsilon), \eta = O(\epsilon^2)$ and $\beta_1 = O(\epsilon^2)$. Compared to [30], our approach maintains a communication complexity of $O(1/\epsilon^3)$, but significantly reduces the sample complexity on each machine from $O(1/\epsilon^6)$ from $O(1/\epsilon^4)$, requiring only a batch size of $O(1)$ rather than a large batch size of $O(1/\epsilon^2)$. Our results match the communication and sample complexity in [17], which tackles a simpler two-level compositional problem and achieved sample complexity of $O(1/\epsilon^4)$ per machine. Considering that the sample complexity for a two-level compositional problem in a centralized setting would be $O(n/\epsilon^4)$ [66], our approach realizes a linear speed-up proportional to the number of machines.

## 6  FGDRO-KL-Adam

In this section, we introduce an adaptive algorithm, FGDRO-KL-Adam, to address the problem (5) with KL regularization, as detailed in Algorithm 3. This algorithm incorporates Adam-type updates at local steps, which have been shown to outperform SGD in centralized settings. While previous studies [59] in federated settings have implemented Adam-type updates at the global step but retained SGD for local updates, which may be sub-optimal.

Similar to Algorithm 2, $u$ and $v$ are used to estimate the local loss and the global function $g(\mathbf{w})$, respectively. The variables $\mathbf{h}$ and $\mathbf{m}$ are updated in a manner consistent with Algorithm 2. Under Assumption 5.1, the behavior of the $u, v, \mathbf{m}$ estimators is addressed as previously discussed.

The primary distinction in Algorithm 3 lies in its adaptive updates for the local model $\mathbf{w}_{i,t}^r$. Here, $\mathbf{m}$ serves a role akin to the first-order momentum in Adam, and we introduce $\mathbf{q}_{i,t}^r$ to estimate the second-order momentum:

$$\mathbf{q}_{i,t}^r = (1 - \beta_4)\mathbf{m}_{i,t-1}^r + \beta_4 \mathbf{h}_{i,t}^r. \tag{8}$$

Subsequently, the local models are updated adaptively using the formula:

$$\mathbf{w}_{i,t}^r = \mathbf{w}_{i,t-1}^r - \eta \frac{\mathbf{m}_{i,t}^r}{\sqrt{\mathbf{q}_{i,t}^r} + \tau}, \tag{9}$$

where both the square root and division are performed element-wise.

A key step in the analysis is to address the coordinate-wise update, as in the following lemma.

**Lemma 6.1.** *Using $L$-smooth of $F$, for some proper constants $C$ and $G$, by setting $\eta = O\left(\frac{1}{\sqrt{RI}}\right)$, $\beta_1 = O\left(\frac{1}{\sqrt{RI}}\right)$, we have*

$$F(\bar{\mathbf{w}}_t^r) \leq F(\bar{\mathbf{w}}_{t-1}^r) + \frac{\eta}{\tau}\|\nabla F(\bar{\mathbf{w}}_{t-1}^r) - \bar{\mathbf{m}}_{t-1}^r\|^2 + \frac{\eta}{\tau}\beta_3^2 C - \frac{\eta}{2(G+\tau)}\|\nabla F(\bar{\mathbf{w}}_{t-1}^r)\|^2. \tag{10}$$

Finally, we show that FGDRO-KL-Adam has same convergence rate as FGDRO-KL.

**Theorem 6.2.** *Under Assumption 5.1, by setting of $\eta = O\left(\frac{1}{\sqrt{RI}}\right)$, $\beta_1 = O\left(\frac{1}{\sqrt{RI}}\right)$, and $I = R^{1/3}$, Algorithm 3 achieves:*

$$\mathbb{E}[\|\nabla F(\tilde{\mathbf{w}})\|^2] \leq O\left(\frac{1}{R^{2/3}}\right). \tag{11}$$

**Remark.** To achieve an $\epsilon$-stationary point, i.e., $\|\nabla F(\tilde{\mathbf{w}})\|^2 \leq \epsilon^2$, we just need to set $R = O(1/\epsilon^3)$, $I = O(1/\epsilon)$, $\eta = O(\epsilon^2)$ and $\beta_1 = O(\epsilon^2)$. The communication and sample complexities are the same as in Theorem 5.3. Our analysis, following the framework in [16], requires $\sqrt{\mathbf{q}_{i,t}^r} + \tau$ to be both upper and lower bounded. It is achieved by the upper bound assumption and choice of $\tau$, ensuring $\tau \leq \sqrt{\mathbf{q}_{i,t}^r} + \tau \leq G + \tau$, which is utilized similarly in [59]. However, Guo et al. [16] did not cover the federated learning scenario or the compositional problems. It is important to note that we have not developed an Adam-type variant for FGDRO-CVaR. This is due to the need for accurate gradient estimation in the analysis of Adam-type updates, which is achieved using the moving estimator $\mathbf{m}$. But in CVaR variant, the nonsmooth nature renders a moving average for subgradient $\partial F(\mathbf{w})$ not provably accurate.

## 7  Experiments

**Datasets and Neural Networks**  We use Pile [15], CivilComments [5], Camelyon17 [1], iWild-Cam2020 [3], and Poverty [74]. For Pile, we preprocess it as [72], for the others, we use the preprocessed version by [37]. We utilized natural data splits where data from the same hospital, web source, location, or demographic group were placed on the same client machine. These experiments have involved with highly imbalanced number of data on clients. Data statistics are presented in the Appendix E. Additiona experiments on Cifar 10 with Dirichlet distributions over 100 clients are reported in Appendix G.

The **Pile** data set is a large language data set. We use the uncopyrighted version [29] which has 17 domains, and each domain is allocated to one machines. We use the GPT2 model [58] as implemented by [28] with 12 hidden layers, 12 attention heads, and 768 embeddings and hidden states. We measure the performance using worst log-perplexity and average log-perplexity of testing groups. **CivilComments** is a toxicity classification of the online comment task in diversified demographic identities. We train on four groups based on the presence of 'Black' and toxicity labels, deploying each on a separate machine, and use the DistilBERT base-uncased model [60] to predict toxicity. We measure the performance using worst group accuracy and average accuracy of testing groups. **Camelyon17** focuses on tumor detection from lymph node images [1], with data from five hospitals split into training (3), validation (1), and testing (1) sets. Training uses three machines, each processing data from one hospital, using DenseNet-121 [26]. The **iWildCam2020** dataset consists of wildlife images from various camera traps [3], the dataset is split into training, validation, and testing segments. We use ResNet50 [21] across all datasets and measure performance via Macro F1 score. The **Poverty** dataset contains geographic data aimed at predicting regional poverty levels [74]. We use ResNet50 [21] for our models and evaluate performance using both the Pearson correlation on the worst-performing region and the average across regions.

Table 2: Experiments on Natural Language Task. PPL is abbrevation of perplexity.

| Datasets | Pile | | CivilComments | |
|---|---|---|---|---|
| Metric | Worst Log-PPL | Average Log-PPL | Worst Acc | Average Acc |
| FedAvg | 8.085($\pm$0.0012) | 6.785 ($\pm$0.0020) | 0.6415 ($\pm$0.0007) | 0.7635 ($\pm$0.0012) |
| SCAFFOLD | 7.975 ($\pm$0.0024) | 6.901 ($\pm$0.0031) | 0.6436 ($\pm$0.0016) | 0.7633 ($\pm$0.0019) |
| FedProx | 8.079 ($\pm$0.0038) | 6.724 ($\pm$0.0046) | 0.6430 ($\pm$0.0021) | 0.7692 ($\pm$0.0025) |
| FedAdam | 7.242 ($\pm$0.0048) | 6.479 ($\pm$0.0037) | 0.6567 ($\pm$0.0023) | 0.7664 ($\pm$0.0020) |
| DRFA | 8.014 ($\pm$0.0053) | 6.702 ($\pm$0.0062) | 0.6327 ($\pm$0.0019) | 0.7413 ($\pm$0.0022) |
| DR-DSGD | 8.023 ($\pm$0.0033) | 6.693 ($\pm$0.0030) | 0.6272 ($\pm$0.0006) | 0.7523 ($\pm$0.0011) |
| FGDRO-CVaR | 8.145 ($\pm$0.0039) | 6.907 ($\pm$0.0046) | 0.6693 ($\pm$0.0015) | 0.7571 ($\pm$0.0012) |
| FGDRO-KL | 7.932 ($\pm$0.0048) | 6.664 ($\pm$0.0051) | **0.6921** ($\pm$0.0016) | **0.7734** ($\pm$0.0020) |
| FGDRO-KL-Adam | **3.608** ($\pm$0.0052) | **2.653** ($\pm$0.0040) | 0.6628 ($\pm$0.0007) | 0.7614 ($\pm$0.0005) |

Table 3: Experiments on Image Classification Task

| Datasets | Camelyon17 | iWildCam2020 | PovertyMap | |
|---|---|---|---|---|
| Metric | Acc | Macro F1 | Worst Pearson | Average Pearson |
| FedAvg | 0.8723 ($\pm$0.0074) | 0.4964 ($\pm$0.0125) | 0.7301 ($\pm$0.0064) | 0.7782 ($\pm$0.0077) |
| SCAFFOLD | 0.8851 ($\pm$0.0063) | 0.4527 ($\pm$0.0331) | 0.7229 ($\pm$0.0049) | 0.7814 ($\pm$0.0042) |
| FedProx | 0.8703 ($\pm$0.0157) | 0.3925 ($\pm$0.0228) | 0.7305 ($\pm$0.0063) | 0.7641 ($\pm$0.0070) |
| FedAdam | **0.9493** ($\pm$0.0122) | 0.3570 ($\pm$0.0203) | 0.7294 ($\pm$0.0058) | **0.8273** ($\pm$0.0041) |
| DRFA | 0.8301 ($\pm$0.0174) | 0.4200 ($\pm$0.0149) | 0.7071($\pm$0.0026) | 0.7665 ($\pm$0.0023) |
| DR-DSGD | 0.9270 ($\pm$0.0095) | 0.3157 ($\pm$0.0227) | 0.7155 ($\pm$0.0063) | 0.7770 ($\pm$0.0059) |
| FGDRO-CVaR | 0.8667 ($\pm$0.0110) | 0.5080 ($\pm$0.0174) | 0.7443 ($\pm$0.0052) | 0.7977 ($\pm$0.0051) |
| FGDRO-KL | 0.9243 ($\pm$0.0129) | **0.5201** ($\pm$0.0239) | 0.7254 ($\pm$0.0066) | 0.7829 ($\pm$0.0062) |
| FGDRO-KL-Adam | 0.9399 ($\pm$0.0154) | 0.4489($\pm$0.0205) | **0.7827** ($\pm$0.0071) | 0.8225 ($\pm$0.0060) |

**Baselines** We compare our algorithms FGDRO-CVaR, FGDRO-KL, and FGDRO-KL-Adam with four baselines: FedAvg [48], SCAFFOLD [34], FedProx[40], FedAdam [59], DRFA[11], and DR-DSGD [30].

We tune the initial step size in [1e-4, 1e-3, 1e-2, 1e-1]. All algorithms set the communication interval $I = 32$ unless otherwise specified. The local mini-batch sizes are set to 32. Experiments are run for 20K local iterations except for Pile, which runs for 200K iterations. The $\beta$ parameters of FGDRO-KL and FGDRO-KL-Adam are tuned in $[0.01, 0.1, 0.2, 0.5]$. For each algorithm, we repeat the experiments 3 times with different random seeds and report the averaged performance. Following [72], our FGDRO algorithms for Pile initially train for 20K iterations to obtain domain weights, which are then fixed during subsequent training phases.

**Results** We report the experimental results for natural language processing in Table 2 and those for computer vision in Table 3. We can see that our methods outperform the baselines in most tasks. Our

approaches improve worst-case performance without hurting average case performance. Furthermore, FGDRO-KL-Adam has demonstrated superior performance compared to FGDRO-KL in most cases.

**Ablation Studies**  Here we present some ablation study to examine some aspects of our algorithm design. First, in Figure 1(a), we vary the communication interval $I$ in experiments on the Camelyon dataset. We can see that both our FGDRO-CVaR and FGDRO-KL-Adam algorithms can tolerate skipping a large number of communications without degrading the performance.

To demonstrate the effect of the local adaptive updates. We develop a LocalAdam algorithm (see Appendix D), which optimizes ERM using our design of using Adam steps in local updates. The results are plotted in Figure 1(b). We can see that the LocalAdam algorithm outperforms FedAdam, which uses SGD in local steps and only uses adaptive steps in global communication rounds.

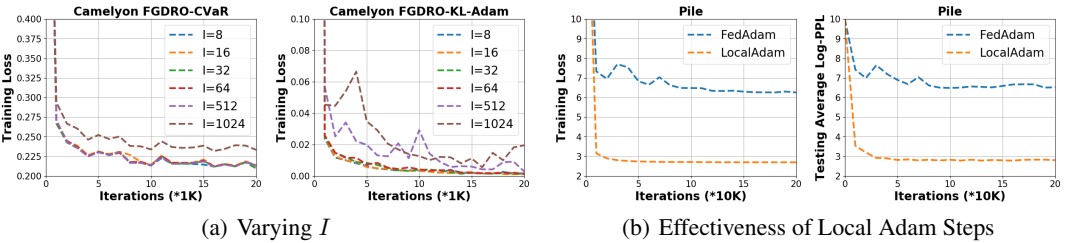

|                     |                                       |
| :-----------------: | :-----------------------------------: |
|   (a) Varying $I$   |  (b) Effectiveness of Local Adam Steps |

Figure 1: Ablation Experiments

# 8    Conclusions

Our algorithm provides a significant advantage in addressing federated group distributionally robust optimization while maintaining low communication and computational complexity. Furthermore, incorporating local adaptive steps has the potential to accelerate the training process beyond the capabilities of traditional approaches that employ SGD in local steps. Various experiments on natural lanugage processing and computer vision have confirmed our theoretical results and underscored the effectiveness of our algorithms. It remains to develop a provable adaptive algorithm for FGDRO-CVaR, which is currently absent due to the non-smoothness and compositional problem structure.

# 9    Impact Statements

This paper is meant to advance the field of federated machine learning. We do not see noticeable negative impact.

## Acknowledgments and Disclosure of Funding

We appreciate the feedback provided by the anonymous reviewers. This work was partially supported by the National Science Foundation Career Award 2246753, the National Science Foundation Award 2246757, 2246756 and 2306572.

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

# A Analysis of FGDRO-CVaR

For non-smooth functions, since it is usually difficult or even impossible to find an $\epsilon$-stationary point, we are interested in finding an $\epsilon$-near stationary point. Following a common technique for finding an $\epsilon$-near stationary point for weakly convex function, we use the Moreau envelope of a general non-smooth $\rho$-weakly-convex $F(\mathbf{x})$ [9], defined as

$$\psi_{(1/\hat{\rho})}(\mathbf{x}) = \min_{\mathbf{x}'}[F(\mathbf{x}') + \frac{\hat{\rho}}{2}\|\mathbf{x}' - \mathbf{x}\|^2].$$

By the properties of Moreau envelop [52, 9], we have that if $\hat{\rho} = 2\rho$, then $\psi_{1/\hat{\rho}}(\cdot)$ is smooth and an $\epsilon$-stationary point of $\psi_{1/\hat{\rho}}(\cdot)$ is an $\epsilon$-near stationary point of $F(\cdot)$.

Since $F(\mathbf{w}, s)$ is non-smooth, we investigate its Moreau envelop, which is

$$\psi_{(1/\hat{\rho})}(\mathbf{w}, s) = \min_{\mathbf{w}', s'}[F(\mathbf{w}', s') + \frac{\hat{\rho}}{2}(\|\mathbf{w}' - \mathbf{w}\|^2 + \|s' - s\|^2)].$$

## A.1 Proof of Lemma 4.2

*Proof.* By updating rule, we have

$$\mathbb{E}\|u_{i,t}^r - g_i(\bar{\mathbf{w}}_t^r)\|^2 = \mathbb{E}\left[\|(1-\beta_1)u_{i,t-1}^r + \beta_1\ell(\mathbf{w}_{i,t-1}^r, \mathbf{z}_{i,t}^r) - g_i(\bar{\mathbf{w}}_t^r)\|^2\right]$$

$$\leq \mathbb{E}\left[\left(1 + \frac{\beta_1}{2}\right)\|(1-\beta_1)u_{i,t-1}^r + \beta_1\ell(\mathbf{w}_{i,t-1}^r, \mathbf{z}_{i,t}^r) - g_i(\bar{\mathbf{w}}_{t-1}^r)\|^2\right] + (1 + \frac{2}{\beta_1})\mathbb{E}\|g_i(\bar{\mathbf{w}}_t^r) - g_i(\bar{\mathbf{w}}_{t-1}^r)\|^2. \tag{12}$$

where
$$\mathbb{E}\|(1-\beta_1)u_{i,t-1}^r + \beta_1\ell(\mathbf{w}_{i,t-1}^r, \mathbf{z}_{i,t}^r) - g_i(\bar{\mathbf{w}}_{t-1}^r)\|^2$$
$$\leq \mathbb{E}\|(1-\beta_1)(u_{i,t-1}^r - g_i(\bar{\mathbf{w}}_{t-1}^r)) + \beta_1(\ell(\bar{\mathbf{w}}_{t-1}^r, \mathbf{z}_{i,t}^r) - g_i(\bar{\mathbf{w}}_{t-1}^r)) + \beta_1(\ell(\mathbf{w}_{i,t-1}^r, \mathbf{z}_{i,t}^r) - \ell(\bar{\mathbf{w}}_{t-1}^r, \mathbf{z}_{i,t}^r))\|^2$$
$$\leq (1 + \frac{\beta_1}{2})\mathbb{E}\|(1-\beta_1)(u_{i,t-1}^r - g_i(\bar{\mathbf{w}}_{t-1}^r)) + \beta_1(\ell(\bar{\mathbf{w}}_{t-1}^r, \mathbf{z}_{i,t}^r) - g_i(\bar{\mathbf{w}}_{t-1}^r))\|^2$$
$$\quad + (1 + \frac{2}{\beta_1})\beta_1^2\|\ell(\mathbf{w}_{i,t-1}^r, \mathbf{z}_{i,t}^r) - \ell(\bar{\mathbf{w}}_{t-1}^r, \mathbf{z}_{i,t}^r)\|^2$$
$$= (1 + \frac{\beta_1}{2})\mathbb{E}\|(1-\beta_1)(u_{i,t-1}^r - g_i(\bar{\mathbf{w}}_{t-1}^r))\|^2 + (1 + \frac{\beta_1}{2})\beta_1^2\|\ell(\bar{\mathbf{w}}_{t-1}^r, \mathbf{z}_{i,t}^r) - g_i(\bar{\mathbf{w}}_{t-1}^r)\|^2$$
$$\quad + (1 + \frac{2}{\beta_1})\beta_1^2\|\ell(\mathbf{w}_{i,t-1}^r, \mathbf{z}_{i,t}^r) - \ell(\bar{\mathbf{w}}_{t-1}^r, \mathbf{z}_{i,t}^r)\|^2, \tag{13}$$

where the last equality uses $\mathbb{E}_{t-1}[\ell(\bar{\mathbf{w}}_{t-1}^r, \mathbf{z}_{i,t}^r) - g_i(\bar{\mathbf{w}}_{t-1}^r)] = 0$.

Since $f(\cdot, \mathbf{z})$ and $g(\cdot, \mathbf{z})$ are Lipschitz and smooth, we know $\|\mathbf{m}_{i,t}^r\|^2 \leq C_g^2$. We also have

$$\|\bar{\mathbf{w}}_t^r - \bar{\mathbf{w}}_{i,t}^r\|^2 = \|(\bar{\mathbf{w}}^r - \eta\frac{1}{N}\sum_{i=1}^N\sum_{t'=1}^t\mathbf{m}_{i,t'}^r) - (\bar{\mathbf{w}}^r - \eta\sum_{t'=1}^t\mathbf{m}_{i,t'}^r)\|^2$$
$$\leq 2\|\frac{1}{N}\sum_{i=1}^N\sum_{t'=1}^t\mathbf{m}_{i,t'}^r\|^2 + 2\|\sum_{t'=1}^t\mathbf{m}_{i,t'}^r\|^2 \leq 4\eta^2 I^2 C_g^2. \tag{14}$$

Therefore,
$$\mathbb{E}\|u_{i,t}^r - g_i(\bar{\mathbf{w}}_t^r)\|^2 \leq (1-\beta_1)\mathbb{E}\|u_{i,t-1}^r - g_i(\bar{\mathbf{w}}_{t-1}^r)\|^2$$
$$\quad + 2\beta_1^2\sigma^2 + 4\beta_1 C_\ell^2\|\bar{\mathbf{w}}_{t-1}^r - \bar{\mathbf{w}}_{i,t-1}^r\|^2 + (1 + \frac{2}{\beta_1})\|g_i(\bar{\mathbf{w}}_t^r) - g_i(\bar{\mathbf{w}}_{t-1}^r)\|^2 \tag{15}$$
$$\leq (1-\beta_1)\mathbb{E}\|u_{i,t-1}^r - g_i(\bar{\mathbf{w}}_{t-1}^r)\|^2 + 2\beta_1^2\sigma^2 + 4\beta_1\eta^2 I^2 C_g^2 + \frac{3}{\beta_1}C_g^2\|\bar{\mathbf{w}}_t^r - \bar{\mathbf{w}}_{t-1}^r\|^2$$

$\square$

## A.2 Proof of Theorem 4.3

*Proof.* Since $f(\cdot)$ is 1-Lipschitz, convex and monitonically nondecreasing, while $\ell(\cdot, \mathbf{z})$ is $C_g$-Lipschitz and $L_g$-smooth, we know that $F(\mathbf{w}, s)$ is $\rho_F := L_g$-weakly convex by

$$
\begin{aligned}
f(g(\mathbf{w}), s) &\geq f(g(\mathbf{w}'), s') + \langle \partial_g(f(g(\mathbf{w}'), s')), g(\mathbf{w}) - g(\mathbf{w}') \rangle + \langle \nabla_g(f(g(\mathbf{w}'), s')), s - s' \rangle \\
&\geq f(g(\mathbf{w}'), s') + \langle \nabla_g(f(g(\mathbf{w}'), s')) \nabla g(\mathbf{w}'), \mathbf{w} - \mathbf{w}' \rangle - \frac{L_g}{2} \|\mathbf{w} - \mathbf{w}'\|^2 \\
&\quad + \langle \nabla_g(f(g(\mathbf{w}'), s')), s - s' \rangle,
\end{aligned}
\tag{16}
$$

where the first inequality uses the convexity of $f(\cdot)$, and the second inequality uses that $\partial f(\cdot) \geq = 0$ and the $L_g$-smoothness of $g(\cdot)$.

With $\hat{\rho} = \max 2\rho_F, 1$, we define

$$
\psi_{(1/\hat{\rho})}(\bar{\mathbf{w}}_t^r, \bar{s}_t^r) = \min_{\mathbf{w}', s'}[F(\mathbf{w}', s') + \frac{\hat{\rho}}{2}(\|\mathbf{w}' - \bar{\mathbf{w}}_t^r\|^2 + \|s' - \bar{s}_t^r\|^2)]
\tag{17}
$$

and

$$
(\hat{\mathbf{w}}_t^r, \hat{s}_t^r) = \arg\min_{\mathbf{w}', s'}[F(\mathbf{w}', s') + \frac{\hat{\rho}}{2}(\|\mathbf{w}' - \bar{\mathbf{w}}_t^r\|^2 + \|s' - \bar{s}_t^r\|^2)],
\tag{18}
$$

then we have the following [9],

$$
\|\hat{\mathbf{w}}_t^r - \bar{\mathbf{w}}_t^r\|_2^2 + \|\hat{s}_t^r - \bar{s}_t^r\|^2 = \frac{1}{\hat{\rho}}\|\nabla\psi_{(1/\hat{\rho})}(\bar{\mathbf{w}}_t^r, \bar{s}_t^r)\|^2.
\tag{19}
$$

Then

$$
\begin{aligned}
\mathbb{E}[\psi_{1/\hat{\rho}}(\bar{\mathbf{w}}_t^r, \bar{s}_t^r)] &= \mathbb{E}\min_{\mathbf{w}', s'}\left[F(\mathbf{w}', s') + \frac{\hat{\rho}}{2}(\|\mathbf{w}' - \bar{\mathbf{w}}_t^r\|^2 + \|s' - \bar{s}_t^r\|^2)\right] \\
&\leq \mathbb{E}\left[F(\hat{\mathbf{w}}_{t-1}^r, \hat{s}_{t-1}^r) + \frac{\hat{\rho}}{2}(\|\hat{\mathbf{w}}_{t-1}^r - \bar{\mathbf{w}}_t^r\|^2 + \|\hat{s}_{t-1}^r - \bar{s}_t^r\|^2)\right] \\
&= \mathbb{E}\left[F(\hat{\mathbf{w}}_{t-1}^r, \hat{s}_{t-1}^r) + \frac{\hat{\rho}}{2}(\|\hat{\mathbf{w}}_{t-1}^r - (\bar{\mathbf{w}}_{t-1}^r - \eta\frac{1}{N}\sum_{i=1}^{N}\mathbf{m}_{i,t}^r)\|^2 + \|\hat{s}_{t-1}^r - (\bar{s}_{t-1}^r - \eta\frac{1}{N}\sum_{i=1}^{N}v_{i,t}^r)\|^2)\right] \\
&\leq \mathbb{E}\left[F(\hat{\mathbf{w}}_{t-1}^r, \hat{s}_{t-1}^r) + \frac{\hat{\rho}}{2}(\|\hat{\mathbf{w}}_{t-1}^r - \bar{\mathbf{w}}_{t-1}^r\|^2 + \|\hat{s}_{t-1}^r - \bar{s}_{t-1}^r\|^2)\right] \\
&\quad + \hat{\rho}\mathbb{E}\left[\eta\langle\hat{\mathbf{w}}_{t-1}^r - \bar{\mathbf{w}}_{t-1}^r, \frac{1}{N}\sum_{i=1}^{N}\mathbf{m}_{i,t}^r\rangle + \eta\langle\hat{s}_{t-1}^r - \bar{s}_{t-1}^r, \frac{1}{N}\sum_{i=1}^{N}v_{i,t}^r\rangle\right] + \hat{\rho}\eta^2 C_g^2,
\end{aligned}
\tag{20}
$$

where we used $\|\mathbf{m}_{i,t}^r\|^2 \leq C_g^2$. Denote $\hat{\mathbf{m}}_t^r = \frac{1}{N}\sum_{i=1}^{N}\partial_u f(u_{i,t}^r, s_{i,t-1}^r)\nabla\ell(\bar{\mathbf{w}}_{i,t-1}^r; \mathbf{z}_{i,t}^r)$, $\hat{v}_t^r = -\frac{1}{N}\sum_{i=1}^{N}\partial_s f(u_{i,t}^r, s_{i,t-1}^r)$, where the sub-differential $\partial_u f(u_{i,t}^r, s_{i,t-1}^r)$ and $\partial_s f(u_{i,t}^r, s_{i,t-1}^r)$ are se-

lected the same in $\mathbf{m}_{i,t}^r$ and $v_{i,t}^r$, respectively. Thus,

$$
\mathbb{E}[\psi_{1/\hat{\rho}}(\bar{\mathbf{w}}_t^r, \bar{s}_t^r)]
$$

$$
\leq \mathbb{E}\bigg[ F(\hat{\mathbf{w}}_{t-1}^r, \hat{s}_{t-1}^r) + \frac{\hat{\rho}}{2}(\|\hat{\mathbf{w}}_{t-1}^r - \bar{\mathbf{w}}_{t-1}^r\|^2 + \|\hat{s}_{t-1}^r - \bar{s}_{t-1}^r\|^2) \bigg]
$$

$$
+ \hat{\rho}\mathbb{E}\bigg[ \eta\langle\hat{\mathbf{w}}_{t-1}^r - \bar{\mathbf{w}}_{t-1}^r, \hat{\mathbf{m}}_t^r\rangle + \eta\langle\hat{s}_{t-1}^r - \bar{s}_{t-1}^r, \hat{v}_t^r\rangle \bigg]
$$

$$
+ \hat{\rho}\mathbb{E}\bigg[ \eta\langle\hat{\mathbf{w}}_{t-1}^r - \bar{\mathbf{w}}_{t-1}^r, \bar{\mathbf{m}}_t^r - \hat{\mathbf{m}}_t^r\rangle + \eta\langle\hat{s}_{t-1}^r - \bar{s}_{t-1}^r, \bar{v}_t^r - \hat{v}_t^r\rangle \bigg] + \hat{\rho}\eta^2 C_g^2
$$

$$
\leq \mathbb{E}\bigg[ F(\hat{\mathbf{w}}_{t-1}^r, \hat{s}_{t-1}^r) + \frac{\hat{\rho}}{2}(\|\hat{\mathbf{w}}_{t-1}^r - \bar{\mathbf{w}}_{t-1}^r\|^2 + \|\hat{s}_{t-1}^r - \bar{s}_{t-1}^r\|^2) \bigg]
$$

$$
+ \hat{\rho}\mathbb{E}\bigg[ \eta\langle\hat{\mathbf{w}}_{t-1}^r - \bar{\mathbf{w}}_{t-1}^r, \hat{\mathbf{m}}_t^r\rangle + \eta\langle\hat{s}_{t-1}^r - \bar{s}_{t-1}^r, \hat{v}_t^r\rangle \bigg]
$$

$$
+ \frac{\eta\hat{\rho}}{2}\|\hat{\mathbf{w}}_{t-1}^r - \bar{\mathbf{w}}_{r-1}^r\|^2 + \frac{\eta\hat{\rho}}{2}\|\hat{s}_{t-1}^r - \bar{s}_{r-1}^r\|^2 + \frac{\eta\hat{\rho}}{2}\|\hat{\mathbf{m}}_t^r - \bar{\mathbf{m}}_t^r\|^2 + \frac{\eta\hat{\rho}}{2}\|\hat{v}_t^r - \bar{v}_t^r\|^2 + \hat{\rho}\eta^2 C_g^2
$$

$$
\tag{21}
$$

Since $f$ is convex, $\partial_u f \geq 0$ and $g_i$ is $\rho_g := L_g$-weakly convex, we have

$$
f(g_i(\hat{\mathbf{w}}_{t-1}^r), \hat{s}_{t-1}^r) - f(u_{i,t}^r, s_{i,t-1}^r)
$$

$$
\geq \partial_u f(u_{i,t}^r, s_{i,t-1}^r)(g_i(\hat{\mathbf{w}}_{t-1}^r) - u_{i,t}^r) + \partial_s f(u_{i,t}^r, \bar{s}_{t-1}^r)(\hat{s}_{t-1}^r - s_{i,t-1}^r)
$$

$$
\geq \partial_u f(u_{i,t}^r, s_{i,t-1}^r)\bigg[ g_i(\bar{\mathbf{w}}_{t-1}^r) - u_{i,t}^r + \langle\nabla g_i(\bar{\mathbf{w}}_{t-1}^r), \hat{\mathbf{w}}_{t-1}^r - \bar{\mathbf{w}}_{t-1}^r\rangle - \frac{\rho_g}{2}\|\hat{\mathbf{w}}_{t-1}^r - \bar{\mathbf{w}}_{t-1}^r\|^2 \bigg]
$$

$$
+ \partial_s f(u_{i,t}^r, s_{i,t-1}^r)(\hat{s}_{t-1}^r - s_{i,t-1}^r).
$$

$$
\tag{22}
$$

Noting $\partial_u f \leq 1$, (22) yields

$$
\langle\hat{\mathbf{m}}_t^r, \hat{\mathbf{w}}_{t-1}^r - \bar{\mathbf{w}}_{t-1}^r\rangle + \langle\hat{v}_t^r, \hat{s}_{t-1}^r - \bar{s}_{t-1}^r\rangle
$$

$$
= \frac{1}{N}\sum_{i=1}^N \langle\partial_u f(u_{i,t}^r, s_{i,t-1}^r)\nabla g_i(\bar{\mathbf{w}}_{t-1}^r), \hat{\mathbf{w}}_{t-1}^r - \bar{\mathbf{w}}_{t-1}^r\rangle + \frac{1}{N}\sum_{i=1}^N \partial_s f(u_{i,t}^r, s_{i,t-1}^r)(\hat{s}_{t-1}^r - s_{i,t-1}^r)
$$

$$
\leq \frac{1}{N}\sum_{i=1}^N \big[ f(g_i(\hat{\mathbf{w}}_{t-1}^r), \hat{s}_{t-1}^r) - f(u_{i,t}^r, s_{i,t-1}^r) - \partial_u f(u_{i,t}^r, s_{i,t-1}^r)[g_i(\bar{\mathbf{w}}_{t-1}^r) - u_{i,t}^r] \big] + \frac{\rho_g}{2}\|\hat{\mathbf{w}}_{t-1}^r - \bar{\mathbf{w}}_{t-1}^r\|^2
$$

$$
\tag{23}
$$

Putting (21) and (23) together, we obtain

$$
\mathbb{E}[\psi_{1/\hat{\rho}}(\bar{\mathbf{w}}_t^r, \bar{s}_t^r)]
$$

$$
\leq \mathbb{E}[\psi_{1/\hat{\rho}}(\bar{\mathbf{w}}_{t-1}^r, \bar{s}_{t-1}^r)] + \frac{\eta\hat{\rho}}{2}\|\hat{\mathbf{w}}_{t-1}^r - \bar{\mathbf{w}}_{r-1}^r\|^2 + \frac{\eta\hat{\rho}}{2}\|\hat{s}_{t-1}^r - \bar{s}_{r-1}^r\|^2 + \frac{\eta\hat{\rho}}{2}\|\hat{\mathbf{m}}_t^r - \bar{\mathbf{m}}_t^r\|^2 + \frac{\eta\hat{\rho}}{2}\|\hat{v}_t^r - \bar{v}_t^r\|^2]
$$

$$
+ \hat{\rho}\eta^2 C_g^2 + \eta\hat{\rho}\frac{1}{N}\sum_{i=1}^N \mathbb{E}\bigg[ f(g_i(\hat{\mathbf{w}}_{t-1}^r), \hat{s}_{t-1}^r) - f(u_{i,t}^r, s_{i,t-1}^r) - \partial_u f(u_{i,t}^r, s_{i,t-1}^r)[g_i(\bar{\mathbf{w}}_{t-1}^r) - u_{i,t}^r]
$$

$$
+ \frac{\rho_g}{2}\|\hat{\mathbf{w}}_{t-1}^r - \bar{\mathbf{w}}_{t-1}^r\|^2 \bigg]
$$

$$
= \mathbb{E}[\psi_{1/\hat{\rho}}(\bar{\mathbf{w}}_{t-1}^r, \bar{s}_{t-1}^r)] + \frac{\eta\hat{\rho}}{2}\|\hat{\mathbf{w}}_{t-1}^r - \bar{\mathbf{w}}_{r-1}^r\|^2 + \frac{\eta\hat{\rho}}{2}\|\hat{s}_{t-1}^r - \bar{s}_{r-1}^r\|^2 + \frac{\eta\hat{\rho}}{2}\|\hat{\mathbf{m}}_t^r - \bar{\mathbf{m}}_t^r\|^2 + \frac{\eta\hat{\rho}}{2}\|\hat{v}_t^r - \bar{v}_t^r\|^2]
$$

$$
+ \hat{\rho}\eta^2 C_g^2 + \eta\hat{\rho}\frac{1}{N}\sum_{i=1}^N \mathbb{E}\bigg[ f(g_i(\hat{\mathbf{w}}_{t-1}^r), \hat{s}_{t-1}^r) - f(g_i(\bar{\mathbf{w}}_{t-1}^r), s_{i,t-1}^r) + f(g_i(\bar{\mathbf{w}}_{t-1}^r), s_{i,t-1}^r) - f(u_{i,t}^r, s_{i,t-1}^r)
$$

$$
- \partial_u f(u_{i,t}^r, s_{i,t-1}^r)[g_i(\bar{\mathbf{w}}_{t-1}^r) - u_{i,t}^r] + \frac{\rho_g}{2}\|\hat{\mathbf{w}}_{t-1}^r - \bar{\mathbf{w}}_{t-1}^r\|^2 \bigg]
$$

$$
\tag{24}
$$

Since $f(g_i(\mathbf{w}), s)$ is $\rho_F$-weakly convex in $\mathbf{w}, s$, $f(g_i(\mathbf{w}), s) + \frac{\hat{\rho}}{2}(\|\mathbf{w} - \bar{\mathbf{w}}_{t-1}^r\|^2 + \|s - \bar{s}_{t-1}^r\|^2)$ is $\hat{\rho} - \rho_F$-strongly convex in $\mathbf{w}, s$. Therefore, we get

$$
\begin{aligned}
&f(g_i(\hat{\mathbf{w}}_{t-1}^r), \hat{s}_{t-1}^r) - f(g_i(\bar{\mathbf{w}}_{t-1}^r), s_{i,t-1}^r) \\
&= [f(g_i(\hat{\mathbf{w}}_{t-1}^r), \hat{s}_{t-1}^r) + \frac{\hat{\rho}}{2}(\|\hat{\mathbf{w}}_{t-1}^r - \bar{\mathbf{w}}_{t-1}^r\|^2 + \|\hat{s}_{t-1}^r - \bar{s}_{t-1}^r\|^2)] \\
&\quad - [f(g_i(\bar{\mathbf{w}}_{t-1}^r), s_{i,t-1}^r) + \frac{\hat{\rho}}{2}(\|\bar{\mathbf{w}}_{t-1}^r - \bar{\mathbf{w}}_{t-1}^r\|^2 + \|s_{i,t-1}^r - \bar{s}_{t-1}^r\|^2)] \\
&\quad - \frac{\hat{\rho}}{2}(\|\hat{\mathbf{w}}_{t-1}^r - \bar{\mathbf{w}}_{t-1}^r\|^2 + \|\hat{s}_{t-1}^r - s_{i,t-1}^r\|^2) + \frac{\hat{\rho}}{2}\|s_{i,t-1}^r - \bar{s}_{t-1}^r\|^2 \\
&\leq (\frac{\rho_F}{2} - \hat{\rho})(\|\hat{\mathbf{w}}_{t-1}^r - \bar{\mathbf{w}}_{t-1}^r\|^2 + \|\hat{s}_{t-1}^r - s_{i,t-1}^r\|^2) + \frac{\hat{\rho}}{2}\|s_{i,t-1}^r - \bar{s}_{t-1}^r\|^2 \\
&\leq -\frac{\rho_F}{2}(\|\hat{\mathbf{w}}_{t-1}^r - \bar{\mathbf{w}}_{t-1}^r\|^2 + \|\hat{s}_{t-1}^r - \bar{s}_{t-1}^r\|^2) + (\hat{\rho} + \rho_F)\|s_{i,t-1}^r - \bar{s}_{t-1}^r\|^2.
\end{aligned}
\tag{25}
$$

Thus,

$$
\begin{aligned}
&\mathbb{E}[\psi_{1/\hat{\rho}}(\bar{\mathbf{w}}_t^r, \bar{s}_t^r)] \\
&\leq \psi_{1/\hat{\rho}}(\bar{\mathbf{w}}_{t-1}^r, \bar{s}_{t-1}^r) + \frac{\eta\hat{\rho}}{2}\|\hat{\mathbf{w}}_{t-1}^r - \bar{\mathbf{w}}_{r-1}^r\|^2 + \frac{\eta\hat{\rho}}{2}\|\hat{s}_{t-1}^r - \bar{s}_{r-1}^r\|^2 + \frac{\eta\hat{\rho}}{2}\|\hat{\mathbf{m}}_t^r - \bar{\mathbf{m}}_t^r\|^2 + \frac{\eta\hat{\rho}}{2}\|\hat{v}_t^r - \bar{v}_t^r\|^2 \\
&\quad + \hat{\rho}\eta^2 C_g^2 + \eta\hat{\rho}\frac{1}{N}\sum_i\left[f(g_i(\hat{\mathbf{w}}_{t-1}^r), \hat{s}_{t-1}^r) - f(g_i(\bar{\mathbf{w}}_{t-1}^r), s_{i,t-1}^r) + f(g_i(\bar{\mathbf{w}}_{t-1}^r), s_{i,t-1}^r) - f(u_{i,t}^r, s_{i,t-1}^r)\right. \\
&\quad \left. - \partial_u f(u_{i,t}^r, s_{i,t-1}^r)[g_i(\bar{\mathbf{w}}_{t-1}^r) - u_{i,t}^r] + \frac{\rho_g}{2}\|\hat{\mathbf{w}}_{t-1}^r - \bar{\mathbf{w}}_{t-1}^r\|^2\right] \\
&\leq \psi_{1/\hat{\rho}}(\bar{\mathbf{w}}_{t-1}^r, \bar{s}_{t-1}^r) + \frac{\eta\hat{\rho}}{2}\|\hat{\mathbf{w}}_{t-1}^r - \bar{\mathbf{w}}_{r-1}^r\|^2 + \frac{\eta\hat{\rho}}{2}\|\hat{s}_{t-1}^r - \bar{s}_{r-1}^r\|^2 + \frac{\eta\hat{\rho}}{2}\|\hat{\mathbf{m}}_t^r - \bar{\mathbf{m}}_t^r\|^2 + \frac{\eta\hat{\rho}}{2}\|\hat{v}_t^r - \bar{v}_t^r\|^2 \\
&\quad + \hat{\rho}\eta^2 C_g^2 + \eta\hat{\rho}(\frac{\rho_F}{2} - \hat{\rho})(\|\hat{\mathbf{w}}_{t-1}^r - \bar{\mathbf{w}}_{t-1}^r\|^2 + \|\hat{s}_{t-1}^r - \bar{s}_{t-1}^r\|^2) + \eta\hat{\rho}(\hat{\rho} + \rho_F)\frac{1}{N}\sum_{i=1}^N\|s_{i,t-1}^r - \bar{s}_{t-1}^r\|^2 \\
&\quad + \eta\hat{\rho}C_f\frac{1}{N}\sum_i\|g_i(\bar{\mathbf{w}}_{t-1}^r) - u_{i,t}^r\| + \eta\hat{\rho}\rho_g\|\hat{\mathbf{w}}_{t-1}^r - \bar{\mathbf{w}}_{t-1}^r\|^2
\end{aligned}
\tag{26}
$$

It follows that

$$
\begin{aligned}
&\mathbb{E}[\psi_{1/\hat{\rho}}(\bar{\mathbf{w}}_t^r, \bar{s}_t^r)] \\
&\leq \psi_{1/\hat{\rho}}(\bar{\mathbf{w}}_{t-1}^r, \bar{s}_{t-1}^r) - \frac{\hat{\rho}}{4}\eta\hat{\rho}(\|\hat{\mathbf{w}}_{t-1}^r - \bar{\mathbf{w}}_{t-1}^r\|^2 + \|\hat{s}_{t-1}^r - \bar{s}_{t-1}^r\|^2) \\
&\quad + \hat{\rho}\eta^2 C_g^2 + \eta\hat{\rho}C_f\frac{1}{N}\sum_i\|g_i(\bar{\mathbf{w}}_{t-1}^r) - u_{i,t}^r\| + \frac{\eta\hat{\rho}}{2}\|\hat{\mathbf{m}}_{t-1}^r - \bar{\mathbf{m}}_{t-1}^r\|^2 + \frac{\eta\hat{\rho}}{2}\|\hat{v}_{t-1}^r - \bar{v}_{t-1}^r\|^2 \\
&\leq \psi_{1/\hat{\rho}}(\bar{\mathbf{w}}_{t-1}^r, \bar{s}_{t-1}^r) - \frac{\eta}{4}\|\nabla\psi_{1/\hat{\rho}}(\bar{\mathbf{w}}_{t-1}^r, \bar{s}_{s-1}^r)\|^2 + \hat{\rho}\eta^2 C_g^2 \\
&\quad + \eta\hat{\rho}C_f\frac{1}{N}\sum_i\|g_i(\bar{\mathbf{w}}_{t-1}^r) - u_{i,t}^r\| + \frac{\eta\hat{\rho}}{2}\eta^2 I^2 C_g^2.
\end{aligned}
\tag{27}
$$

$$
\begin{aligned}
&\frac{1}{RI}\sum_{r=1}^R\sum_{t=1}^I\mathbb{E}\|\nabla\psi_{1/\hat{\rho}}(\bar{\mathbf{w}}_{t-1}^r, \bar{s}_{s-1}^r)\|^2 \leq \\
&\frac{\psi_{1/\hat{\rho}}(\bar{\mathbf{w}}_0^r, \bar{s}_0^r)}{\eta RI} + \eta\hat{\rho}C_g^2 + \frac{1}{NRI}\sum_{r=1}^R\sum_{t=1}^I\sum_{i=1}^N\mathbb{E}\|g_i(\bar{\mathbf{w}}_{t-1}^r) - u_{i,t}^r\| + \eta^2 I^2 C_g^2.
\end{aligned}
\tag{28}
$$

Hence, taking telescoping sum over Lemma 4.2 sum we have

$$
\frac{1}{RIN} \sum_{r=1}^{R} \sum_{t=1}^{I} \sum_{i=1}^{N} \mathbb{E}\|(u_{i,t-1}^{r} - g_i(\bar{\mathbf{w}}_{t-1}^{r}))\|^2
$$
$$
\leq \frac{1}{\beta_1 RIN} \sum_{i=1}^{N} \mathbb{E}\|(u_{i,0}^{0} - g_i(\bar{\mathbf{w}}_0^0))\|^2) + 2\beta_1\sigma^2 + \eta^2 I^2 C_g^2 + \frac{\eta^2}{\beta_1^2} C_g^2.
$$

(29)

Thus,

$$
\mathbb{E}\|\nabla\psi_{1/\hat{\rho}}(\tilde{\mathbf{w}}, \tilde{s})\|^2 = \frac{1}{RI} \sum_{r=1}^{R} \sum_{t=1}^{I} \mathbb{E}\|\nabla\psi_{1/\hat{\rho}}(\tilde{\mathbf{w}}, \tilde{s})\|^2 \leq O\left(\frac{1}{\eta RI} + \eta + \frac{1}{\beta_1 RI} + \sqrt{\beta_1} + \eta I + \frac{\eta}{\beta_1}\right).
$$

(30)

With

$$
(\hat{\mathbf{w}}, \hat{s}) = \arg\min_{\mathbf{w}', s'}[F(\mathbf{w}', s') + \frac{\hat{\rho}}{2}(\|\mathbf{w}' - \tilde{\mathbf{w}}\|^2 + \|s' - \tilde{s}\|^2)],
$$

(31)

We also have that $\|\hat{\mathbf{w}} - \tilde{\mathbf{w}}\|^2 + \|\hat{s} - \tilde{s}\|^2 = \hat{\rho}\|\nabla\psi_{1/\hat{\rho}}(\tilde{\mathbf{w}}, \tilde{s})\|^2$, $|dist(\mathbf{0}, \partial F(\hat{\mathbf{w}}, \hat{s}))|^2 \leq \|\nabla\psi_{1/\hat{\rho}}(\tilde{\mathbf{w}}, \tilde{s})\|^2$ [9]. We can conclude by setting parameters as in the theorem. □

# B  Analysis of FGDRO-KL

## B.1  Lemmas

The behavior of $u$, which is an estimator of $\nabla g_i(\mathbf{w})$'s, is given in the following lemma.

**Lemma B.1.** *Under Assumption 5.1, with some constant $G$, by setting $\eta = O\left(\frac{1}{\sqrt{RI}}\right)$, $\beta_1 = O\left(\frac{1}{\sqrt{RI}}\right)$, Algorithm 2 ensures that*

$$
\mathbb{E}\|u_{i,t}^{r} - \ell(\bar{\mathbf{w}}_t^r; \mathcal{D}_i)\|^2 \leq (1 - \frac{\beta_1}{2})\mathbb{E}\|u_{i,t-1}^{r} - \ell(\bar{\mathbf{w}}_{t-1}^r; \mathcal{D}_i)\|^2 + 3\beta_1\eta^2\beta_3^2 I^4
$$
$$
+ 12\beta_1 C_\ell^2 \eta^2 I \sum_{\tau=0}^{t-1} \mathbb{E}\|\bar{\mathbf{m}}_\tau^r\|^2 + \beta_1^2\sigma^2 + \frac{3}{\beta_1} C_\ell^2 \eta^2 \mathbb{E}\|\bar{\mathbf{m}}_t^r\|^2.
$$

(32)

The behavior of $v$, which is an estimator of $\frac{1}{N} \sum_i g_i(\mathbf{w})$, is given in the following lemma.

**Lemma B.2.** *Under Assumption 5.1, with some constant $C_1$, by setting $\eta = O\left(\frac{1}{\sqrt{RI}}\right)$, $\beta_1 = O\left(\frac{1}{\sqrt{RI}}\right)$, Algorithm 2 ensures that*

$$
\mathbb{E}\|\bar{v}_t^r - g(\bar{\mathbf{w}}_t^r)\|^2 \leq (1 - \beta_2)\mathbb{E}\|\bar{v}_{t-1}^r - g(\bar{\mathbf{w}}_{t-1}^r)\|^2
$$
$$
+ 3\beta_2 \frac{1}{N} \sum_{i=1}^{N} C_1 \mathbb{E}\|u_{i,t}^r - \ell(\mathbf{w}; \mathcal{D}_i)\|^2 + \frac{3}{\beta_2} C_g^2 \mathbb{E}\|\bar{\mathbf{w}}_t^r - \bar{\mathbf{w}}_{t-1}^r\|^2.
$$

(33)

## B.2 Proof of Lemma B.1

*Proof.* Denoting $\bar{\mathbf{w}}_t^r = \frac{1}{N} \sum_{i=1}^{N} \bar{\mathbf{w}}_{i,t}^r$, we have $\bar{\mathbf{w}}_t^r = \bar{\mathbf{w}}_{t-1}^r - \eta \bar{\mathbf{m}}_t^r$, and

$$
\begin{aligned}
&\mathbb{E}\|u_{i,t}^r - \ell(\bar{\mathbf{w}}_t^r; \mathcal{D}_i)\|^2 = \mathbb{E}\|(1-\beta_1)u_{i,t-1}^r + \beta_1 \ell(\mathbf{w}_{i,t-1}^r; \mathbf{z}_{i,t}^r) - \ell(\bar{\mathbf{w}}_t^r; \mathcal{D}_i)\|^2 \\
&= \mathbb{E}\|(1-\beta_1)(u_{i,t-1}^r - \ell(\bar{\mathbf{w}}_{t-1}^r; \mathcal{D}_i)) + \beta_1 (\ell(\mathbf{w}_{i,t-1}^r; \mathbf{z}_{i,t}^r) - \ell(\bar{\mathbf{w}}_{t-1}^r; \mathcal{D}_i)) + (\ell(\bar{\mathbf{w}}_{t-1}^r; \mathcal{D}_i) - \ell(\bar{\mathbf{w}}_t^r; \mathcal{D}_i))\|^2 \\
&\leq \mathbb{E}\left(1 + \frac{\beta_1}{2}\right) \|(1-\beta_1)(u_{i,t-1}^r - \ell(\bar{\mathbf{w}}_{t-1}^r; \mathcal{D}_i)) + \beta_1 (\ell(\mathbf{w}_{i,t-1}^r; \mathbf{z}_{i,t}^r) - \ell(\bar{\mathbf{w}}_{t-1}^r; \mathcal{D}_i))\|^2 \\
&\quad + \mathbb{E}\left(1 + \frac{2}{\beta_1}\right) \|\ell(\bar{\mathbf{w}}_{t-1}^r; \mathcal{D}_i) - \ell(\bar{\mathbf{w}}_t^r; \mathcal{D}_i)\|^2 \\
&\leq \left(1 + \frac{\beta_1}{2}\right) \|(1-\beta_1)(u_{i,t-1}^r - \ell(\bar{\mathbf{w}}_{t-1}^r; \mathcal{D}_i)) + \beta_1(\ell(\mathbf{w}_{i,t-1}^r; \mathbf{z}_{i,t}^r) - \ell(\mathbf{w}_{i,t-1}^r; \mathcal{D}_i)) \\
&\qquad\qquad + \beta_1 (\ell(\mathbf{w}_{i,t-1}^r; \mathcal{D}_i) - \ell(\bar{\mathbf{w}}_{t-1}^r; \mathcal{D}_i))\|^2 + \frac{3}{\beta_1} C_\ell^2 \|\bar{\mathbf{w}}_{t-1}^r - \bar{\mathbf{w}}_t^r\|^2 \\
&\overset{(a)}{=} \left(1 + \frac{\beta_1}{2}\right) \mathbb{E}\|(1-\beta_1)(u_{i,t-1}^r - \ell(\bar{\mathbf{w}}_{t-1}^r; \mathcal{D}_i)) + \beta_1 (\ell(\mathbf{w}_{i,t-1}^r; \mathcal{D}_i) - \ell(\bar{\mathbf{w}}_{t-1}^r; \mathcal{D}_i))\|^2 \\
&\quad + \beta_1^2 \mathbb{E}\|\ell(\mathbf{w}_{i,t-1}^r; \mathbf{z}_{i,t}^r) - \ell(\mathbf{w}_{i,t-1}^r; \mathcal{D}_i)\|^2 + \frac{3}{\beta_1} C_\ell^2 \mathbb{E}\|\bar{\mathbf{w}}_{t-1}^r - \bar{\mathbf{w}}_t^r\|^2 \\
&\leq \left(1 + \frac{\beta_1}{2}\right)^2 (1-\beta_1)^2 \mathbb{E}\|u_{i,t-1}^r - \ell(\bar{\mathbf{w}}_{t-1}^r; \mathcal{D}_i)\|^2 + \left(1 + \frac{2}{\beta_1}\right) \beta_1^2 \mathbb{E}\|\ell(\mathbf{w}_{i,t-1}^r; \mathcal{D}_i) - \ell(\bar{\mathbf{w}}_{t-1}^r; \mathcal{D}_i)\|^2 \\
&\quad + \beta_1^2 \sigma^2 + \frac{3}{\beta_1} C_\ell^2 \eta^2 \mathbb{E}\|\bar{\mathbf{m}}_t^r\|^2,
\end{aligned}
$$

$$(34)$$

where (a) holds due to the fact that $\mathbb{E}_{t-1}[\ell(\mathbf{w}_{i,t}^r; \mathbf{z}_{i,t}^r) - \ell(\mathbf{w}_{i,t}^r; \mathcal{D}_i)] = 0$. Moreover, we have

$$
\begin{aligned}
&\mathbb{E}\|\ell(\mathbf{w}_{i,t-1}^r; \mathcal{D}_i) - \ell(\bar{\mathbf{w}}_{t-1}^r; \mathcal{D}_i)\|^2 \\
&\leq C_\ell^2 \mathbb{E}\|\mathbf{w}_{i,t-1}^r - \bar{\mathbf{w}}_{t-1}^r\|^2 \\
&\leq 2C_\ell^2 \mathbb{E}\|\mathbf{w}_{i,t-1}^r - \bar{\mathbf{w}}^r\|^2 + 2C_\ell^2 \mathbb{E}\|\bar{\mathbf{w}}^r - \bar{\mathbf{w}}_{t-1}^r\|^2 \\
&\leq 2C_\ell^2 \eta^2 \mathbb{E}\|\sum_{\tau=1}^{t-1} \mathbf{m}_{i,\tau}^r\|^2 + 2C_\ell^2 \eta^2 \mathbb{E}\|\frac{1}{N} \sum_{k=1}^{N} \sum_{\tau=1}^{t-1} \mathbf{m}_{k,\tau}^r\|^2 \\
&\leq 4C_\ell^2 \eta^2 I \sum_{\tau=1}^{t-1} \mathbb{E}\|\mathbf{m}_{i,\tau}^r - \bar{\mathbf{m}}_\tau^r\|^2 + 6C_\ell^2 \eta^2 I \sum_{\tau=1}^{t-1} \mathbb{E}\|\bar{\mathbf{m}}_\tau^r\|^2,
\end{aligned}
$$

$$(35)$$

and

$$
\begin{aligned}
\mathbb{E}\|\mathbf{m}_{i,\tau}^r - \bar{\mathbf{m}}_\tau^r\|^2 &= \mathbb{E}\left\| \left( \sum_{\tau=1}^{t} (1-\beta_3)^{\tau-1} \beta_3 \frac{1}{v_{i,\tau}^r} g(u_{i,\tau}^r) \nabla \ell(\mathbf{w}_{i,\tau-1}^r; \mathbf{z}_{\tau,t}^r) + (1-\beta_3)^\tau \bar{\mathbf{m}}_0^r \right) \right. \\
&\qquad \left. - \left( \sum_{\tau=1}^{t} (1-\beta_3)^{\tau-1} \beta_3 \frac{1}{N} \sum_{k=1}^{N} \frac{1}{v_{k,t}^r} g(u_{k,t}^r) \nabla \ell(\mathbf{w}_{k,t-1}^r; \mathbf{z}_{k,t}^r) + (1-\beta_3)^\tau \bar{\mathbf{m}}_0^r \right) \right\|^2 \\
&\leq 2\mathbb{E}\| \sum_{\tau=1}^{t} (1-\beta_3)^{\tau-1} \beta_3 \frac{1}{v_{i,\tau}^r} g(u_{i,\tau}^r) \nabla \ell(\mathbf{w}_{i,\tau-1}^r; \mathbf{z}_{\tau,t}^r)\|^2 \\
&\quad + 2\mathbb{E}\| \sum_{\tau=1}^{t} (1-\beta_3)^{\tau-1} \beta_3 \frac{1}{N} \sum_{k=1}^{N} \frac{1}{v_{k,t}^r} g(u_{k,t}^r) \nabla \ell(\mathbf{w}_{k,t-1}^r; \mathbf{z}_{k,t}^r)\|^2,
\end{aligned}
$$

$$(36)$$

which yields

$$\mathbb{E}\|\mathbf{m}_{i,\tau}^r - \bar{\mathbf{m}}_\tau^r\|^2$$

$$\leq 2t \sum_{\tau=1}^t \beta_3^2 \mathbb{E}[\|\frac{1}{v_{i,\tau}^r}\|^2 \|g(u_{i,\tau}^r)\|^2 \|\nabla\ell(\mathbf{w}_{i,\tau-1}^r; \mathbf{z}_{i,\tau}^r)\|^2]$$

$$+ 2t \sum_{\tau=1}^t \beta_3^2 \frac{1}{N} \sum_{k=1}^N \mathbb{E}[\|\frac{1}{v_{k,t}^r}\|^2 \|g(u_{k,\tau}^r)\|^2 \|\nabla\ell(\mathbf{w}_{k,t-1}^r; \mathbf{z}_{i,\tau}^r)\|^2]$$

$$\leq 8I^2 \beta_3^2 C_1^2 \bigg( \mathbb{E}\|\nabla\ell(\mathbf{w}_{i,\tau-1}^r; \mathbf{z}_{i,\tau}^r) - \nabla\ell(\mathbf{w}_{i,\tau-1}^r; \mathcal{D}_i)\|^2 + \mathbb{E}\|\nabla\ell(\mathbf{w}_{i,\tau-1}^r; \mathcal{D}_i)\|^2$$

$$+ \frac{1}{N} \sum_{k=1}^N (\mathbb{E}\|\nabla\ell(\mathbf{w}_{k,\tau-1}^r; \mathbf{z}_{k,\tau}^r) - \nabla\ell(\mathbf{w}_{k,\tau-1}^r; \mathcal{D}_k)\|^2 + \mathbb{E}\|\nabla\ell(\mathbf{w}_{k,\tau-1}^r; \mathcal{D}_k)\|^2) \bigg)$$

$$\leq 16I^2 \beta_3^2 C_1^2 (\sigma^2 + C_\ell^2),$$

$$(37)$$

with $C_1 = \exp(C_0/\lambda)$. Thus,

$$\mathbb{E}\|u_{i,t}^r - \ell(\bar{\mathbf{w}}_t^r; \mathcal{D}_i)\|^2 \leq (1 - \frac{\beta_1}{2})\mathbb{E}\|u_{i,t-1}^r - \ell(\bar{\mathbf{w}}_{t-1}^r; \mathcal{D}_i)\|^2 + 3\beta_1 \eta^2 \beta_3^2 I^4$$

$$+ 12\beta_1 C_\ell^2 \eta^2 I \sum_{\tau=0}^{t-1} \mathbb{E}\|\bar{\mathbf{m}}_\tau^r\|^2 + \beta_1^2 \sigma^2 + \frac{3}{\beta_1} C_\ell^2 \eta^2 \mathbb{E}\|\bar{\mathbf{m}}_t^r\|^2,$$

$$(38)$$

with $C_2 = 288(C_\ell^2 C_1^2(\sigma^2 + C_\ell^2))$.

$\square$

## B.3 Proof of Lemma B.2

*Proof.* We have

$$\|\bar{v}_t^r - g(\bar{\mathbf{w}}_t^r)\|^2 = \|(1 - \beta_2)\bar{v}_{t-1}^r + \beta_2 \frac{1}{N} \sum_{i=1}^N \exp(u_{i,t}^r/\lambda) - g(\bar{\mathbf{w}}_t^r)\|^2$$

$$= \left\| (1 - \beta_2)(\bar{v}_{t-1}^r - g(\bar{\mathbf{w}}_{t-1}^r)) + \beta_2 \left( \frac{1}{N} \sum_{i=1}^N \exp(u_{i,t}^r/\lambda) - g(\bar{\mathbf{w}}_{t-1}^r) \right) - g(\bar{\mathbf{w}}_t^r) + g(\bar{\mathbf{w}}_{t-1}^r) \right\|^2$$

$$\leq \left(1 + \frac{\beta_2}{2}\right) \left\| (1 - \beta_2)(\bar{v}_{t-1}^r - g(\bar{\mathbf{w}}_{t-1}^r)) + \beta_2 \left( \frac{1}{N} \sum_{i=1}^N \exp(u_{i,t}^r/\lambda) - g(\bar{\mathbf{w}}_{t-1}^r) \right) \right\|^2$$

$$+ \left(1 + \frac{2}{\beta_2}\right) \|g(\bar{\mathbf{w}}_t^r) - g(\bar{\mathbf{w}}_{t-1}^r)\|^2$$

$$\leq \left(1 + \frac{\beta_2}{2}\right)^2 (1 - \beta_2)^2 \|\bar{v}_{t-1}^r - g(\bar{\mathbf{w}}_{t-1}^r)\|^2 + \left(1 + \frac{2}{\beta_2}\right) \beta_2^2 \left\| \frac{1}{N} \sum_{i=1}^N [\exp(u_{i,t}^r/\lambda) - g_i(\bar{\mathbf{w}}_{t-1}^r)] \right\|^2$$

$$+ \frac{3}{\beta_2} C_g^2 \|\bar{\mathbf{w}}_t^r - \bar{\mathbf{w}}_{t-1}^r\|^2$$

$$\leq (1 - \beta_2)\|\bar{v}_{t-1}^r - g(\bar{\mathbf{w}}_{t-1}^r)\|^2 + 3\beta_2 \frac{1}{N} \sum_{i=1}^N C_1 \|u_{i,t}^r - \ell(\mathbf{w}; \mathcal{D}_i)\|^2 + \frac{3}{\beta_2} C_g^2 \|\bar{\mathbf{w}}_t^r - \bar{\mathbf{w}}_{t-1}^r\|^2,$$

$$(39)$$

where $C_1 = \exp(C_0/\lambda)$.

$\square$

## B.4 Proof of Lemma 5.2

*Proof.* Here we analyze the $\bar{\mathbf{m}}$, which is the moving average estimator of the gradient,

$$\|\bar{\mathbf{m}}_t^r - \nabla F(\bar{\mathbf{w}}_t^r)\|^2 = \left\|(1-\beta_3)\bar{\mathbf{m}}_{t-1}^r + \beta_3 \frac{1}{N}\sum_{i=1}^{N}\frac{1}{v_{i,t}^r}g(u_{i,t}^r)\nabla\ell(\mathbf{w}_{i,t-1}^r;\mathbf{z}_{i,t}^r) - \nabla F(\bar{\mathbf{w}}_t^r)\right\|^2$$

$$\leq \left\|(1-\beta_3)(\bar{\mathbf{m}}_{t-1}^r - \nabla F(\bar{\mathbf{w}}_{t-1}^r)) + \beta_3\Big(\frac{1}{N}\sum_{i=1}^{N}\frac{1}{v_{i,t}^r}\exp(u_{i,t}^r/\lambda)\nabla\ell(\mathbf{w}_{i,t-1}^r;\mathbf{z}_{i,t}^r) - \nabla F(\bar{\mathbf{w}}_{t-1}^r)\Big)\right.$$
$$\left. + \nabla F(\bar{\mathbf{w}}_{t-1}^r) - \nabla F(\bar{\mathbf{w}}_t^r)\right\|^2$$

$$\leq \left(1+\frac{\beta_3}{2}\right)\underbrace{\left\|(1-\beta_3)(\bar{\mathbf{m}}_{t-1}^r - \nabla F(\bar{\mathbf{w}}_{t-1}^r)) + \beta_3\Big(\frac{1}{N}\sum_{i=1}^{N}\frac{1}{v_{i,t}^r}\exp(u_{i,t}^r/\lambda)\nabla\ell(\mathbf{w}_{i,t-1}^r;\mathbf{z}_{i,t}^r) - \nabla F(\bar{\mathbf{w}}_{t-1}^r)\Big)\right\|^2}_{(A)}$$

$$+ \left(1+\frac{2}{\beta_3}\right)\|\nabla F(\bar{\mathbf{w}}_{t-1}^r) - \nabla F(\bar{\mathbf{w}}_t^r)\|^2,$$

(40)

where

$$(A) = \left\|(1-\beta_3)(\bar{\mathbf{m}}_{t-1}^r - \nabla F(\bar{\mathbf{w}}_{t-1}^r))\right.$$
$$+ \beta_3\left(\frac{1}{N}\sum_{i=1}^{N}\frac{1}{v_{i,t}^r}\exp(u_{i,t}^r/\lambda)\nabla\ell(\mathbf{w}_{i,t-1}^r;\mathbf{z}_{i,t}^r) - \frac{1}{N}\sum_{i=1}^{N}\frac{1}{v_{i,t}^r}\exp(u_{i,t-1}^r/\lambda)\nabla\ell(\mathbf{w}_{i,t-1}^r;\mathbf{z}_{i,t}^r)\right)$$
$$+ \beta_3\left(\frac{1}{N}\sum_{i=1}^{N}\frac{1}{v_{i,t}^r}\exp(u_{i,t-1}^r/\lambda)\nabla\ell(\mathbf{w}_{i,t-1}^r;\mathbf{z}_{i,t}^r) - \frac{1}{N}\sum_{i=1}^{N}\frac{1}{v_{i,t}^r}\exp(u_{i,t-1}^r/\lambda)\nabla\ell(\mathbf{w}_{i,t-1}^r;\mathcal{D}_i)\right)$$
$$+ \beta_3\left(\frac{1}{N}\sum_{i=1}^{N}\frac{1}{v_{i,t}^r}\exp(u_{i,t-1}^r/\lambda)\nabla\ell(\mathbf{w}_{i,t-1}^r;\mathcal{D}_i) - \frac{1}{N}\sum_{i=1}^{N}\frac{1}{v_{i,t}^r}\exp(\ell(\bar{\mathbf{w}}_{t-1}^r;\mathcal{D}_i)/\lambda)\nabla\ell(\mathbf{w}_{i,t-1}^r;\mathcal{D}_i)\right)$$
$$+ \beta_3\left(\frac{1}{N}\sum_{i=1}^{N}\frac{1}{v_{i,t}^r}\exp(\ell(\bar{\mathbf{w}}_{t-1}^r;\mathcal{D}_i)/\lambda)\nabla\ell(\mathbf{w}_{i,t-1}^r;\mathcal{D}_i) - \frac{1}{N}\sum_{i=1}^{N}\frac{1}{\bar{v}_t^r}\exp(\ell(\bar{\mathbf{w}}_{t-1}^r;\mathcal{D}_i)/\lambda)\nabla\ell(\bar{\mathbf{w}}_{t-1}^r;\mathcal{D}_i)\right)$$
$$+ \left.\beta_3\left(\frac{1}{N}\sum_{i=1}^{N}\frac{1}{\bar{v}_t^r}\exp(\ell(\bar{\mathbf{w}}_{t-1}^r;\mathcal{D}_i)/\lambda)\nabla\ell(\bar{\mathbf{w}}_{t-1}^r;\mathcal{D}_i) - \nabla F(\bar{\mathbf{w}}_{t-1}^r)\right)\right\|^2,$$

$$\leq \left(1+\frac{\beta_3}{2}\right)\left\|(1-\beta_3)(\bar{\mathbf{m}}_{t-1}^r - \nabla F(\bar{\mathbf{w}}_{t-1}^r))\right.$$
$$+ \beta_3\left(\frac{1}{N}\sum_{i=1}^{N}\frac{1}{v_{i,t}^r}\exp(u_{i,t-1}^r/\lambda)\nabla\ell(\mathbf{w}_{i,t-1}^r;\mathbf{z}_{i,t}^r) - \frac{1}{N}\sum_{i=1}^{N}\frac{1}{v_{i,t}^r}\exp(u_{i,t-1}^r/\lambda)\nabla\ell(\mathbf{w}_{i,t-1}^r;\mathcal{D}_i)\right)$$
$$+ \beta_3\left(\frac{1}{N}\sum_{i=1}^{N}\frac{1}{v_{i,t}^r}\exp(u_{i,t-1}^r/\lambda)\nabla\ell(\mathbf{w}_{i,t-1}^r;\mathcal{D}_i) - \frac{1}{N}\sum_{i=1}^{N}\frac{1}{v_{i,t}^r}\exp(\ell(\bar{\mathbf{w}}_{t-1}^r;\mathcal{D}_i)/\lambda)\nabla\ell(\mathbf{w}_{i,t-1}^r;\mathcal{D}_i)\right)$$
$$+ \beta_3\left(\frac{1}{N}\sum_{i=1}^{N}\frac{1}{v_{i,t}^r}\exp(\ell(\bar{\mathbf{w}}_{t-1}^r;\mathcal{D}_i)/\lambda)\nabla\ell(\mathbf{w}_{i,t-1}^r;\mathcal{D}_i) - \frac{1}{N}\sum_{i=1}^{N}\frac{1}{\bar{v}_t^r}\exp(\ell(\bar{\mathbf{w}}_{t-1}^r;\mathcal{D}_i)/\lambda)\nabla\ell(\bar{\mathbf{w}}_{t-1}^r;\mathcal{D}_i)\right)$$
$$+ \left.\beta_3\left(\frac{1}{N}\sum_{i=1}^{N}\frac{1}{\bar{v}_t^r}\exp(\ell(\bar{\mathbf{w}}_{t-1}^r;\mathcal{D}_i)/\lambda)\nabla\ell(\bar{\mathbf{w}}_{t-1}^r;\mathcal{D}_i) - \nabla F(\bar{\mathbf{w}}_{t-1}^r)\right)\right\|^2,$$
$$+ (1+\frac{2}{\beta_3})\beta_3^2\left\|\frac{1}{N}\sum_{i=1}^{N}\frac{1}{v_{i,t}^r}\exp(u_{i,t}^r/\lambda)\nabla\ell(\mathbf{w}_{i,t-1}^r;\mathbf{z}_{i,t}^r) - \frac{1}{N}\sum_{i=1}^{N}\frac{1}{v_{i,t}^r}\exp(u_{i,t-1}^r/\lambda)\nabla\ell(\mathbf{w}_{i,t-1}^r;\mathbf{z}_{i,t}^r)\right\|^2,$$

(41)

which is followed by

$$
\begin{aligned}
(A) &\leq \left(1 + \frac{\beta_3}{2}\right) \bigg\| (1 - \beta_3)(\bar{\mathbf{m}}_{t-1}^r - \nabla F(\bar{\mathbf{w}}_{t-1}^r)) \\
&\quad + \beta_3 \left(\frac{1}{N} \sum_{i=1}^{N} \frac{1}{v_{i,t}^r} \exp(u_{i,t-1}^r/\lambda) \nabla \ell(\mathbf{w}_{i,t-1}^r; \mathcal{D}_i) - \frac{1}{N} \sum_{i=1}^{N} \frac{1}{v_{i,t}^r} \exp(\ell(\bar{\mathbf{w}}_{t-1}^r; \mathcal{D}_i)/\lambda) \nabla \ell(\mathbf{w}_{i,t-1}^r; \mathcal{D}_i)\right) \\
&\quad + \beta_3 \left(\frac{1}{N} \sum_{i=1}^{N} \frac{1}{v_{i,t}^r} \exp(\ell(\bar{\mathbf{w}}_{t-1}^r; \mathcal{D}_i)/\lambda) \nabla \ell(\mathbf{w}_{i,t-1}^r; \mathcal{D}_i) - \frac{1}{N} \sum_{i=1}^{N} \frac{1}{\bar{v}_t^r} \exp(\ell(\bar{\mathbf{w}}_{t-1}^r; \mathcal{D}_i)/\lambda) \nabla \ell(\bar{\mathbf{w}}_{t-1}^r; \mathcal{D}_i)\right) \\
&\quad + \beta_3 \left(\frac{1}{N} \sum_{i=1}^{N} \frac{1}{\bar{v}_t^r} \exp(\ell(\bar{\mathbf{w}}_{t-1}^r; \mathcal{D}_i)/\lambda) \nabla \ell(\bar{\mathbf{w}}_{t-1}^r; \mathcal{D}_i) - \nabla F(\bar{\mathbf{w}}_{t-1}^r)\right) \bigg\|^2, \\
&\quad + \left(1 + \frac{\beta_3}{2}\right) \bigg\| \beta_3 \left(\frac{1}{N} \sum_{i=1}^{N} \frac{1}{v_{i,t}^r} \exp(u_{i,t-1}^r/\lambda) \nabla \ell(\mathbf{w}_{i,t-1}^r; \mathbf{z}_{i,t}^r) - \frac{1}{N} \sum_{i=1}^{N} \frac{1}{v_{i,t}^r} \exp(u_{i,t-1}^r/\lambda) \nabla \ell(\mathbf{w}_{i,t-1}^r; \mathcal{D}_i)\right) \bigg\|^2 \\
&\quad + (1 + \frac{2}{\beta_3}) \beta_3 \bigg\| \frac{1}{N} \sum_{i=1}^{N} \frac{1}{v_{i,t}^r} \exp(u_{i,t}^r/\lambda) \nabla \ell(\mathbf{w}_{i,t-1}^r; \mathbf{z}_{i,t}^r) - \frac{1}{N} \sum_{i=1}^{N} \frac{1}{v_{i,t}^r} \exp(u_{i,t-1}^r/\lambda) \nabla \ell(\mathbf{w}_{i,t-1}^r; \mathbf{z}_{i,t}^r) \bigg\|^2.
\end{aligned}
\tag{42}
$$

Then it leads to

$$
\begin{aligned}
\|\bar{\mathbf{m}}_t^r &- \nabla F(\bar{\mathbf{w}}_t^r)\|^2 \\
&\leq \left(1 + \frac{\beta_3}{2}\right)^2 \bigg\| (1 - \beta_3)(\bar{\mathbf{m}}_{t-1}^r - \nabla F(\bar{\mathbf{w}}_{t-1}^r)) \\
&\quad + \beta_3 \left(\frac{1}{N} \sum_{i=1}^{N} \frac{1}{v_{i,t}^r} \exp(u_{i,t-1}^r/\lambda) \nabla \ell(\mathbf{w}_{i,t-1}^r; \mathcal{D}_i) - \frac{1}{N} \sum_{i=1}^{N} \frac{1}{v_{i,t}^r} \exp(\ell(\bar{\mathbf{w}}_{t-1}^r; \mathcal{D}_i)/\lambda) \nabla \ell(\mathbf{w}_{i,t-1}^r; \mathcal{D}_i)\right) \\
&\quad + \beta_3 \left(\frac{1}{N} \sum_{i=1}^{N} \frac{1}{v_{i,t}^r} \exp(\ell(\bar{\mathbf{w}}_{t-1}^r; \mathcal{D}_i)/\lambda) \nabla \ell(\mathbf{w}_{i,t-1}^r; \mathcal{D}_i) - \frac{1}{N} \sum_{i=1}^{N} \frac{1}{\bar{v}_t^r} \exp(\ell(\bar{\mathbf{w}}_{t-1}^r; \mathcal{D}_i)/\lambda) \nabla \ell(\bar{\mathbf{w}}_{t-1}^r; \mathcal{D}_i)\right) \\
&\quad + \beta_3 \left(\frac{1}{N} \sum_{i=1}^{N} \frac{1}{\bar{v}_t^r} \exp(\ell(\bar{\mathbf{w}}_{t-1}^r; \mathcal{D}_i)/\lambda) \nabla \ell(\bar{\mathbf{w}}_{t-1}^r; \mathcal{D}_i) - \nabla F(\bar{\mathbf{w}}_{t-1}^r)\right) \bigg\|^2 \\
&\quad + (1 + \frac{\beta_3}{2})^2 \beta_3^2 \bigg\| \frac{1}{N} \sum_{i=1}^{N} \frac{1}{v_{i,t}^r} \exp(u_{i,t-1}^r/\lambda) \nabla \ell(\mathbf{w}_{i,t-1}^r; \mathbf{z}_{i,t}^r) - \frac{1}{N} \sum_{i=1}^{N} \frac{1}{v_{i,t}^r} \exp(u_{i,t-1}^r/\lambda) \nabla \ell(\mathbf{w}_{i,t-1}^r; \mathcal{D}_i) \bigg\|^2 \\
&\quad + \frac{6}{\beta_3} \beta_3^2 \bigg\| \frac{1}{N} \sum_{i=1}^{N} \frac{1}{v_{i,t}^r} \exp(u_{i,t}^r/\lambda) \nabla \ell(\mathbf{w}_{i,t-1}^r; \mathbf{z}_{i,t}^r) - \frac{1}{N} \sum_{i=1}^{N} \frac{1}{v_{i,t}^r} \exp(u_{i,t-1}^r/\lambda) \nabla \ell(\mathbf{w}_{i,t-1}^r; \mathbf{z}_{i,t}^r) \bigg\|^2 \\
&\quad + \left(1 + \frac{2}{\beta_3}\right) \|\nabla F(\bar{\mathbf{w}}_{t-1}^r) - \nabla F(\bar{\mathbf{w}}_t^r)\|^2.
\end{aligned}
\tag{43}
$$

Thus, it follows that

$$\|\bar{\mathbf{m}}_t^r - \nabla F(\bar{\mathbf{w}}_t^r)\|^2 \leq \left(1 + \frac{\beta_3}{2}\right)^2 (1-\beta_3)^3 \|\bar{\mathbf{m}}_{t-1}^r - \nabla F(\bar{\mathbf{w}}_{t-1}^r)\|^2$$

$$+ \frac{8}{\beta_3}\beta_3^2 \left\| \frac{1}{N}\sum_{i=1}^N \frac{1}{v_{i,t}^r} \exp(u_{i,t-1}^r/\lambda)\nabla\ell(\mathbf{w}_{i,t-1}^r;\mathcal{D}_i) - \frac{1}{N}\sum_{i=1}^N \frac{1}{v_{i,t}^r}\exp(\ell(\bar{\mathbf{w}}_{t-1}^r;\mathcal{D}_i)/\lambda)\nabla\ell(\mathbf{w}_{i,t-1}^r;\mathcal{D}_i) \right\|^2 \qquad ①$$

$$+ \frac{8}{\beta_3}\beta_3^2 \left\| \frac{1}{N}\sum_{i=1}^N \frac{1}{v_{i,t}^r}\exp(\ell(\bar{\mathbf{w}}_{t-1}^r;\mathcal{D}_i)/\lambda)\nabla\ell(\mathbf{w}_{i,t-1}^r;\mathcal{D}_i) - \frac{1}{N}\sum_{i=1}^N \frac{1}{\bar{v}_t^r}\exp(\ell(\bar{\mathbf{w}}_t^r;\mathcal{D}_i)/\lambda)\nabla\ell(\bar{\mathbf{w}}_{t-1}^r;\mathcal{D}_i) \right\|^2 \qquad ②$$

$$+ \frac{8}{\beta_3}\beta_3^2 \left\| \frac{1}{N}\sum_{i=1}^N \frac{1}{\bar{v}_t^r}\exp(\ell(\bar{\mathbf{w}}_{t-1}^r;\mathcal{D}_i)/\lambda)\nabla\ell(\mathbf{w}_{i,t-1}^r;\mathcal{D}_i) - \nabla F(\bar{\mathbf{w}}_{t-1}^r) \right\|^2 \qquad ③$$

$$+ \beta_3^2 \left\| \frac{1}{N}\sum_{i=1}^N \frac{1}{v_{i,t}^r}\exp(u_{i,t-1}^r/\lambda)\nabla\ell(\mathbf{w}_{i,t-1}^r;\mathbf{z}_{i,t}^r) - \frac{1}{N}\sum_{i=1}^N \frac{1}{v_{i,t}^r}\exp(u_{i,t-1}^r/\lambda)\nabla\ell(\mathbf{w}_{i,t-1}^r;\mathcal{D}_i) \right\|^2 \qquad ④$$

$$+ \frac{8}{\beta_3}\beta_3^2 \left\| \frac{1}{N}\sum_{i=1}^N \frac{1}{v_{i,t}^r}\exp(u_{i,t}^r/\lambda)\nabla\ell(\mathbf{w}_{i,t-1}^r;\mathbf{z}_{i,t}^r) - \frac{1}{N}\sum_{i=1}^N \frac{1}{v_{i,t}^r}\exp(u_{i,t-1}^r/\lambda)\nabla\ell(\mathbf{w}_{i,t-1}^r;\mathbf{z}_{i,t}^r) \right\|^2 \qquad ⑤$$

$$+ \left(1 + \frac{2}{\beta_3}\right)\|\nabla F(\bar{\mathbf{w}}_{t-1}^r) - \nabla F(\bar{\mathbf{w}}_t^r)\|^2 \qquad ⑥.$$

$$(44)$$

We address each term as follows. ① can be bounded as

$$\mathbb{E}\left[ 8\beta_3 \left\| \frac{1}{N}\sum_{i=1}^N \frac{1}{v_{i,t}^r}\exp(u_{i,t-1}^r/\lambda)\nabla\ell(\mathbf{w}_{i,t-1}^r;\mathcal{D}_i) - \frac{1}{N}\sum_{i=1}^N \frac{1}{v_{i,t}^r}\exp(\ell(\bar{\mathbf{w}}_{t-1}^r;\mathcal{D}_i)/\lambda)\nabla\ell(\mathbf{w}_{i,t-1}^r;\mathcal{D}_i) \right\|^2 \right]$$

$$\leq \beta_3 \frac{1}{N}\sum_{i=1}^N C_\ell^2 C_1^2 \|u_{i,t-1}^r - \ell(\bar{\mathbf{w}}_{t-1}^r;\mathcal{D}_i)\|^2.$$

$$(45)$$

② can be bounded as

$$\mathbb{E}\left[ 8\beta_3 \left\| \frac{1}{N}\sum_{i=1}^N \frac{1}{v_{i,t}^r}\exp(\ell(\bar{\mathbf{w}}_{t-1}^r;\mathcal{D}_i)/\lambda)\nabla\ell(\mathbf{w}_{i,t-1}^r;\mathcal{D}_i) - \frac{1}{N}\sum_{i=1}^N \frac{1}{\bar{v}_t^r}\exp(\ell(\bar{\mathbf{w}}_t^r;\mathcal{D}_i)/\lambda)\nabla\ell(\mathbf{w}_{i,t-1}^r;\mathcal{D}_i) \right\|^2 \right]$$

$$\leq \mathbb{E}\left[ 16\beta_3 \left\| \frac{1}{N}\sum_{i=1}^N \frac{1}{v_{i,t}^r}\exp(\ell(\bar{\mathbf{w}}_{t-1}^r;\mathcal{D}_i)/\lambda)\nabla\ell(\mathbf{w}_{i,t-1}^r;\mathcal{D}_i) - \frac{1}{N}\sum_{i=1}^N \frac{1}{v_{i,t}^r}\exp(\ell(\bar{\mathbf{w}}_t^r;\mathcal{D}_i)/\lambda)\nabla\ell(\mathbf{w}_{i,t-1}^r;\mathcal{D}_i) \right\|^2 \right]$$

$$+ \mathbb{E}\left[ 16\beta_3 \left\| \frac{1}{N}\sum_{i=1}^N \frac{1}{v_{i,t}^r}\exp(\ell(\bar{\mathbf{w}}_t^r;\mathcal{D}_i)/\lambda)\nabla\ell(\mathbf{w}_{i,t-1}^r;\mathcal{D}_i) - \frac{1}{N}\sum_{i=1}^N \frac{1}{\bar{v}_t^r}\exp(\ell(\bar{\mathbf{w}}_t^r;\mathcal{D}_i)/\lambda)\nabla\ell(\mathbf{w}_{i,t-1}^r;\mathcal{D}_i) \right\|^2 \right]$$

$$\leq 16\beta_3 C_\ell^2 C_1^2 \mathbb{E}\|\bar{\mathbf{w}}_{t-1}^r - \bar{\mathbf{w}}_t^r\|^2 + 16\beta_3 \frac{1}{N}\sum_{i=1}^N C_\ell^2 C_1^2 \|v_{i,t}^r - \bar{v}_t^r\|^2$$

$$\leq 16\beta_3 C_\ell^2 C_1^2 \eta^2 \mathbb{E}\|\bar{\mathbf{m}}_t^r\|^2 + 16\beta_3 \frac{1}{N}\sum_{i=1}^N C_\ell^2 C_1^2 \|v_{i,t}^r - \bar{v}_t^r\|^2.$$

$$(46)$$

③ can be bounded as

$$\mathbb{E}\left[8\beta_3\left\|\frac{1}{N}\sum_{i=1}^{N}\frac{1}{\bar{v}_t^r}\exp(\ell(\bar{\mathbf{w}}_{t-1}^r;\mathcal{D}_i)/\lambda)\nabla\ell(\mathbf{w}_{i,t-1}^r;\mathcal{D}_i)-\nabla F(\bar{\mathbf{w}}_{t-1}^r)\right\|^2\right]$$

$$=\mathbb{E}\left[16\beta_3\left\|\frac{1}{N}\sum_{i=1}^{N}\frac{1}{\bar{v}_t^r}\exp(\ell(\bar{\mathbf{w}}_{t-1}^r;\mathcal{D}_i)/\lambda)\nabla\ell(\mathbf{w}_{i,t-1}^r;\mathcal{D}_i)-\frac{1}{N}\sum_{i=1}^{N}\frac{1}{g(\bar{\mathbf{w}}_{t-1}^r)}\exp(\ell(\bar{\mathbf{w}}_{t-1}^r;\mathcal{D}_i)/\lambda)\nabla\ell(\mathbf{w}_{i,t-1}^r;\mathcal{D}_i)\right\|\right]$$

$$+\mathbb{E}\left[16\beta_3\left\|\frac{1}{N}\sum_{i=1}^{N}\frac{1}{g(\bar{\mathbf{w}}_t^r)}\exp(\ell(\bar{\mathbf{w}}_{t-1}^r;\mathcal{D}_i)/\lambda)\nabla\ell(\mathbf{w}_{i,t-1}^r;\mathcal{D}_i)-\frac{1}{N}\sum_{i=1}^{N}\frac{1}{g(\bar{\mathbf{w}}_t^r)}\exp(\ell(\bar{\mathbf{w}}_{t-1}^r;\mathcal{D}_i)/\lambda)\nabla\ell(\bar{\mathbf{w}}_t^r;\mathcal{D}_i)\right\|\right]$$

$$\leq 32\beta_3 C_1^2 C_\ell^2\|\bar{v}_t^r-g(\bar{\mathbf{w}}_t^r)\|^2+32\beta_3 C_1^2 C_\ell^2 C_g^2\|\bar{\mathbf{w}}_{t-1}^r-\bar{\mathbf{w}}_t^r\|^2+16\beta_3 C_1^2 L_\ell^2\frac{1}{N}\sum_{i=1}^{N}\|\mathbf{w}_{i,t-1}^r-\bar{\mathbf{w}}_{t-1}^r\|^2$$

$$\overset{(a)}{\leq}32\beta_3 C_1^2 C_\ell^2\|\bar{v}_t^r-g(\bar{\mathbf{w}}_t^r)\|^2+32\beta_3 C_1^2 C_\ell^2 C_g^2\eta^2\|\bar{\mathbf{m}}_t^r\|^2$$

$$+64C_\ell^2\eta^4 I^2\sum_{\tau=1}^{t-1}\mathbb{E}\|\mathbf{m}_{i,\tau}^r-\bar{\mathbf{m}}_\tau^r\|^2+96C_\ell^2\eta^2 I\sum_{\tau=1}^{t-1}\mathbb{E}\|\bar{\mathbf{m}}_\tau^r\|^2,$$

$$(47)$$

where (a) follows from (35). ④ can be bounded as

$$\beta_3^2\mathbb{E}\left\|\frac{1}{N}\sum_{i=1}^{N}\frac{1}{v_{i,t}^r}\exp(u_{i,t-1}^r/\lambda)\nabla\ell(\mathbf{w}_{i,t-1}^r;\mathbf{z}_{i,t}^r)-\frac{1}{N}\sum_{i=1}^{N}\frac{1}{v_{i,t}^r}\exp(u_{i,t-1}^r/\lambda)\nabla\ell(\mathbf{w}_{i,t-1}^r;\mathcal{D}_i)\right\|^2$$

$$=\beta_3^2\frac{1}{N^2}\sum_{i=1}^{N}\mathbb{E}\left\|\frac{1}{v_{i,t}^r}\exp(u_{i,t-1}^r/\lambda)\nabla\ell(\mathbf{w}_{i,t-1}^r;\mathbf{z}_{i,t}^r)-\frac{1}{N}\sum_{i=1}^{N}\frac{1}{v_{i,t}^r}\exp(u_{i,t-1}^r/\lambda)\nabla\ell(\mathbf{w}_{i,t-1}^r;\mathcal{D}_i)\right\|^2$$

$$\leq\beta_3^2 C_1^2\frac{\sigma^2}{N}.$$

$$(48)$$

as machines are independent conditioned on iteration $t-1$.

⑤ can be bounded as

$$\mathbb{E}\left[8\beta_3\left\|\frac{1}{N}\sum_{i=1}^{N}\frac{1}{v_{i,t}^r}\exp(u_{i,t}^r/\lambda)\nabla\ell(\mathbf{w}_{i,t-1}^r;\mathbf{z}_{i,t}^r)-\frac{1}{N}\sum_{i=1}^{N}\frac{1}{v_{i,t}^r}\exp(u_{i,t-1}^r/\lambda)\nabla\ell(\mathbf{w}_{i,t-1}^r;\mathbf{z}_{i,t}^r)\right\|^2\right]$$

$$\leq\mathbb{E}\left[8\beta_3\frac{1}{N}\sum_{i=1}^{N}C_2^2\left\|\exp(u_{i,t}^r/\lambda)\nabla\ell(\mathbf{w}_{i,t-1}^r;\mathbf{z}_{i,t}^r)-\exp(u_{i,t-1}^r/\lambda)\nabla\ell(\mathbf{w}_{i,t-1}^r;\mathbf{z}_{i,t}^r)\right\|^2\right]$$

$$\leq\mathbb{E}\left[24\beta_3\frac{1}{N}\sum_{i=1}^{N}C_2^2\left\|\exp(u_{i,t}^r/\lambda)\nabla\ell(\mathbf{w}_{i,t-1}^r;\mathbf{z}_{i,t}^r)-\exp(u_{i,t}^r/\lambda)\nabla\ell(\mathbf{w}_{i,t-1}^r;\mathcal{D}_i)\right\|^2\right]$$

$$+\mathbb{E}\left[24\beta_3\frac{1}{N}\sum_{i=1}^{N}C_2^2\left\|\exp(u_{i,t-1}^r/\lambda)\nabla\ell(\mathbf{w}_{i,t-1}^r;\mathbf{z}_{i,t}^r)-\exp(u_{i,t-1}^r/\lambda)\nabla\ell(\mathbf{w}_{i,t-1}^r;\mathcal{D}_i)\right\|^2\right]$$

$$+\mathbb{E}\left[24\beta_3\frac{1}{N}\sum_{i=1}^{N}C_2^2\left\|\exp(u_{i,t}^r/\lambda)\nabla\ell(\mathbf{w}_{i,t-1}^r;\mathcal{D}_i)-\exp(u_{i,t-1}^r/\lambda)\nabla\ell(\mathbf{w}_{i,t-1}^r;\mathcal{D}_i)\right\|^2\right]$$

$$\leq 48\beta_3 C_2^2 C_1^2\sigma^2+\mathbb{E}\left[24\beta_3\frac{1}{N}\sum_{i=1}^{N}L_\ell^2 C_1^2\|u_{i,t}^r-u_{i,t-1}^r\|^2\right]$$

$$\leq 48\beta_3 C_2^2 C_1^2\sigma^2+24\beta_3 L_\ell^2 C_1^2 C_0^2\beta_1^2,$$

$$(49)$$

where the first inequality uses $v_{i,t}^r\geq 1$ as $\ell(\cdot)\geq 0$.

$F(\mathbf{w})$ is $L_F := C_f L_g + C_g^2 L_f$-smooth. With $\eta \le \beta_3/(3L_F^2)$, ⑥ can be bounded as

$$
\begin{aligned}
\left(1 + \frac{2}{\beta_3}\right) \mathbb{E}\|\nabla F(\bar{\mathbf{w}}_{t-1}^r) - \nabla F(\bar{\mathbf{w}}_t^r)\|^2 &\le \frac{3}{\beta_3} L_F^2 \mathbb{E}\|\bar{\mathbf{w}}_{t-1}^r - \bar{\mathbf{w}}_t^r\|^2 = \frac{3}{\beta_3} L_F^2 \eta^2 \mathbb{E}\|\bar{\mathbf{m}}_t^r\|^2 \\
&\le \eta \mathbb{E}\|\bar{\mathbf{m}}_t^r - \bar{\mathbf{m}}_{t-1}^r + \bar{\mathbf{m}}_{t-1}^r - \nabla F(\bar{\mathbf{w}}_{t-1}^r) + \nabla F(\bar{\mathbf{w}}_{t-1}^r)\|^2 \\
&\le 3\eta \mathbb{E}\|\bar{\mathbf{m}}_t^r - \bar{\mathbf{m}}_{t-1}^r\|^2 + 3\eta\|\bar{\mathbf{m}}_{t-1}^r - \nabla F(\bar{\mathbf{w}}_{t-1}^r)\|^2 + 3\eta \mathbb{E}\|\nabla F(\bar{\mathbf{w}}_{t-1}^r)\|^2 \\
&\overset{(a)}{\le} 3\eta(4\beta_3^2 \mathbb{E}\|\bar{\mathbf{m}}_{t-1}^r - \nabla F(\bar{\mathbf{w}}_{t-1}^r)\|^2 + 4\beta_3^2 \mathbb{E}\|\nabla F(\bar{\mathbf{m}}_{t-1}^r)\|^2 + 4\beta_3^2 C_1^2(C_\ell^2 + \sigma^2)) \\
&\quad + 3\eta \mathbb{E}\|\bar{\mathbf{m}}_{t-1}^r - \nabla F(\bar{\mathbf{w}}_{t-1}^r)\|^2 + 3\eta \mathbb{E}\|\nabla F(\bar{\mathbf{w}}_{t-1}^r)\|^2 \\
&\le 4\eta \mathbb{E}\|\bar{\mathbf{m}}_{t-1}^r - \nabla F(\bar{\mathbf{w}}_{t-1}^r)\|^2 + 4\eta \mathbb{E}\|\nabla F(\bar{\mathbf{w}}_{t-1}^r)\|^2 + 4\beta_3^2 C_1^2(C_\ell^2 + \sigma^2))
\end{aligned}
\tag{50}
$$

where (a) uses

$$
\begin{aligned}
\mathbb{E}\|\bar{\mathbf{m}}_t^r - \bar{\mathbf{m}}_{t-1}^r\|^2 &= \mathbb{E}\|(1 - \beta_3)\bar{\mathbf{m}}_{t-1}^r + \beta_3 \frac{1}{N}\sum_{i=1}^N \frac{1}{v_{i,t}^r} g(u_{i,t}^r) \nabla\ell(\mathbf{w}_{i,t-1}^r; \mathbf{z}_{i,t}^r) - \bar{\mathbf{m}}_{t-1}^r\|^2 \\
&\le 2\beta_3^2 \mathbb{E}\|\bar{\mathbf{m}}_{t-1}^r\|^2 + 2\beta_3^2 \mathbb{E}\|\frac{1}{N}\sum_{i=1}^N \frac{1}{v_{i,t}^r} g(u_{i,t}^r) \nabla\ell(\mathbf{w}_{i,t-1}^r; \mathbf{z}_{i,t}^r)\|^2 \\
&\le 4\beta_3^2 \mathbb{E}\|\bar{\mathbf{m}}_{t-1}^r - \nabla F(\bar{\mathbf{w}}_{t-1}^r)\|^2 + 4\beta_3^2 \mathbb{E}\|\nabla F(\bar{\mathbf{m}}_{t-1}^r)\|^2 + 4\beta_3^2 C_1^2(C_\ell^2 + \sigma^2).
\end{aligned}
\tag{51}
$$

Thus,

$$
\begin{aligned}
\|\bar{\mathbf{m}}_t^r - \nabla F(\bar{\mathbf{w}}_t^r)\|^2 &\le (1 - \frac{\beta_3}{2})\|\bar{\mathbf{m}}_{t-1}^r - \nabla F(\bar{\mathbf{w}}_{t-1}^r)\|^2 + \beta_3 \frac{1}{N}\sum_{i=1}^N C_\ell^2 C_1^2 \|u_{i,t-1}^r - \ell(\bar{\mathbf{w}}_{t-1}^r; \mathcal{D}_i)\|^2 \\
&\quad + 16\beta_3 C_\ell^2 C_1^2 \eta^2 \mathbb{E}\|\bar{\mathbf{m}}_t^r\|^2 + 16\beta_3 \frac{1}{N}\sum_{i=1}^N C_\ell^2 C_1^2 \|v_{i,t}^r - \bar{v}_t^r\|^2 \\
&\quad + 32\beta_3 C_1^2 C_\ell^2 \|\bar{v}_t^r - g(\bar{\mathbf{w}}_t^r)\|^2 + 32\beta_3 C_1^2 C_\ell^2 C_g^2 \eta^2 \|\bar{\mathbf{m}}_t^r\|^2 \\
&\quad + 64 C_\ell^2 \eta^4 I^2 \sum_{\tau=1}^{t-1} \mathbb{E}\|\mathbf{m}_{i,\tau}^r - \bar{\mathbf{m}}_\tau^r\|^2 + 96 C_\ell^2 \eta^2 I \sum_{\tau=1}^{t-1} \mathbb{E}\|\bar{\mathbf{m}}_\tau^r\|^2 + \beta_3^2 C_1^2 \frac{\sigma^2}{N} \\
&\quad + 48\beta_3 C_2^2 C_1^2 \sigma^2 + 24\beta_3 L_\ell^2 C_1^2 C_0^2 \beta_1^2 \\
&\quad + 4\eta \mathbb{E}\|\bar{\mathbf{m}}_{t-1}^r - \nabla F(\bar{\mathbf{w}}_{t-1}^r)\|^2 + 4\eta \mathbb{E}\|\nabla F(\bar{\mathbf{w}}_{t-1}^r)\|^2 + 4\beta_3^2 C_1^2(C_\ell^2 + \sigma^2))
\end{aligned}
\tag{52}
$$

We conclude the proof by setting the parameters as in the Lemma. $\qquad\square$

*Proof of Theorem 5.3.* Using Lemma 5.2, with $\beta_2 = O(\beta_3)$, we have

$$
\begin{aligned}
\frac{1}{RI}\sum_{r=1}^R \sum_{t=1}^I \frac{\beta_3}{2} \mathbb{E}\|\bar{\mathbf{m}}_{t-1}^0 - \nabla F(\bar{\mathbf{w}}_{t-1}^0)\|^2 &\le O\Bigg(\frac{\mathbb{E}\|\bar{\mathbf{m}}_0^r - \nabla F(\bar{\mathbf{w}}_0^r)\|^2}{RI} + 2\beta_3^3 I^2 G^2 + \beta_3^2 C_2 \frac{\sigma^2}{N} \\
&\quad + \frac{1}{RKN}\sum_{i=1}^N \mathbb{E}\|u_{i,0}^0 - \nabla F(\bar{\mathbf{w}}_0^0)\|^2 + \frac{1}{RI}\sum_{r=1}^R \sum_{t=1}^I \eta \mathbb{E}\|\nabla F(\bar{\mathbf{w}}_{t-1}^r)\|^2 + \beta_1^2 \sigma^2\Bigg).
\end{aligned}
\tag{53}
$$

Using $L_F$-smooth of $F$, we have

$$F(\bar{\mathbf{w}}_t^r) \leq F(\bar{\mathbf{w}}_{t-1}^r) + \nabla F(\bar{\mathbf{w}}_{t-1}^r)^\top (\mathbf{w}_t^r - \bar{\mathbf{w}}_{t-1}^r) + \frac{L_F}{2}\|\bar{\mathbf{w}}_t^r - \bar{\mathbf{w}}_{t-1}^r\|^2$$

$$= F(\bar{\mathbf{w}}_{t-1}^r) - \eta \nabla F(\bar{\mathbf{w}}_{t-1}^r)^\top \bar{\mathbf{m}}_t^r + \frac{L_F}{2}\eta^2\|\bar{\mathbf{m}}_t^r\|^2$$

$$\leq F(\bar{\mathbf{w}}_{t-1}^r) - \eta \nabla F(\bar{\mathbf{w}}_{t-1}^r)^\top (\bar{\mathbf{m}}_t^r - \nabla F(\bar{\mathbf{w}}_{t-1}^r) + \nabla F(\bar{\mathbf{w}}_{t-1}^r))$$
$$+ L_F\eta^2(\|\bar{\mathbf{m}}_t^r - \nabla F(\bar{\mathbf{w}}_{t-1}^r)\|^2 + \|\nabla F(\bar{\mathbf{w}}_{t-1}^r)\|^2) \tag{54}$$

$$\leq F(\bar{\mathbf{w}}_{t-1}^r) - \eta\|\nabla F(\bar{\mathbf{w}}_{t-1}^r)\|^2 - \eta \nabla F(\bar{\mathbf{w}}_{t-1}^r)^\top (\bar{\mathbf{m}}_t^r - \nabla F(\bar{\mathbf{w}}_{t-1}^r))$$
$$+ \frac{\eta}{4}(\|\bar{\mathbf{m}}_t^r - \nabla F(\bar{\mathbf{w}}_{t-1}^r)\|^2 + \|\nabla F(\bar{\mathbf{w}}_{t-1}^r)\|^2)$$

$$\leq F(\bar{\mathbf{w}}_{t-1}^r) - \frac{\eta}{4}\|\nabla F(\bar{\mathbf{w}}_{t-1}^r)\|^2 + \frac{3\eta}{4}\|\bar{\mathbf{m}}_t^r - \nabla F(\bar{\mathbf{w}}_{t-1}^r))\|^2.$$

Therefore,

$$\mathbb{E}\left[\|\nabla F(\tilde{\mathbf{w}})\|^2\right] = \mathbb{E}\left[\frac{1}{RI}\sum_{r=1}^{R}\sum_{t=1}^{I}\|\nabla F(\bar{\mathbf{w}}_{t-1}^r)\|^2\right] \leq O\left(\frac{1}{\eta RI} + \beta_1\sigma^2 + \beta_3^2 I^2 + \eta L_F\right). \tag{55}$$

We conclude the proof by setting the parameters as the theorem.

$\square$

## C    Analysis of FGDRO-KL-Adam

*Proof of Theorem 6.2.* Lemma B.1, Lemma B.2 and Lemma 5.2 still hold. Specifically, denoting $\bar{\mathbf{w}}_t^r = \frac{1}{N}\sum_{i=1}^{N}\bar{\mathbf{w}}_{i,t}^r$, we have

$$\mathbb{E}\|u_{i,t}^r - \ell(\bar{\mathbf{w}}_t^r; \mathcal{D}_i)\|^2$$

$$\leq (1-\beta_1)\|u_{i,t-1}^r - \ell(\bar{\mathbf{w}}_{t-1}^r; \mathcal{D}_i)\|^2 + 6\beta_1\eta^2 I^2 C_\ell^2 G^2 + \beta_1^2\sigma^2 + \frac{\eta^2}{\beta_1}C_2\|\bar{\mathbf{m}}_t^r - \nabla F(\bar{\mathbf{w}}_t^r)\|^2 \tag{56}$$

$$+ \frac{\eta^2}{\beta_1}C_2\|\nabla F(\bar{\mathbf{w}}_t^r)\|^2$$

where (a) holds due to the fact that $\mathbb{E}_{t-1}[\ell(\mathbf{w}_{i,t}^r; \mathbf{z}_{i,t}^r) - \ell(\mathbf{w}_{i,t}^r; \mathcal{D}_i)] = 0$.

And we have

$$\|\bar{v}_t^r - g(\bar{\mathbf{w}}_t^r)\|^2 \leq (1-\beta_2)\|\bar{v}_{t-1}^r - g(\bar{\mathbf{w}}_{t-1}^r)\|^2 + 3\beta_2\frac{1}{N}\sum_{i=1}^{N}C_1\|u_{i,t}^r - \ell(\mathbf{w}; \mathcal{D}_i)\|^2 + \frac{3}{\beta_2}C_g^2\|\bar{\mathbf{w}}_t^r - \bar{\mathbf{w}}_{t-1}^r\|^2 \tag{57}$$

where $C_1 = \exp(C_0/\lambda)$.

Moreover, we have

$$\frac{1}{RI}\sum_{r=1}^{R}\sum_{t=1}^{I}\frac{\beta_3}{2}\mathbb{E}\|\bar{\mathbf{m}}_{t-1}^r - \nabla F(\bar{\mathbf{w}}_{t-1}^r)\|^2 \leq O\left(\frac{\mathbb{E}\|\bar{\mathbf{m}}_0^r - \nabla F(\bar{\mathbf{w}}_0^r)\|^2}{RI} + 2\beta_3^3 I^2 G^2 + \beta_3^2 C_2\frac{\sigma^2}{N}\right.$$

$$\left. + \frac{1}{RKN}\sum_{i=1}^{N}\mathbb{E}\|u_{i,0}^0 - \nabla F(\bar{\mathbf{w}}_0^0)\|^2 + \frac{1}{RI}\sum_{r=1}^{R}\sum_{t=1}^{I}\eta\mathbb{E}\|\nabla F(\bar{\mathbf{w}}_{t-1}^r)\|^2 + \beta_1^2\sigma^2\right). \tag{58}$$

Using $L_F$-smooth of $F$, we have

$$F(\bar{\mathbf{w}}_t^r) \leq F(\bar{\mathbf{w}}_{t-1}^r) + \nabla F(\bar{\mathbf{w}}_{t-1}^r)^\top (\bar{\mathbf{w}}_t^r - \bar{\mathbf{w}}_{t-1}^r) + \frac{L}{2}\|\bar{\mathbf{w}}_t^r - \bar{\mathbf{w}}_{t-1}^r\|^2$$

$$\leq F(\bar{\mathbf{w}}_{t-1}^r) - \nabla F(\bar{\mathbf{w}}_{t-1}^r)^\top (\tilde{\eta}_t \circ \bar{\mathbf{m}}_t^r) + \frac{L}{2}\|\tilde{\eta}^2 \circ \bar{\mathbf{m}}_t^r\|^2$$

$$= F(\bar{\mathbf{w}}_{t-1}^r) + \frac{1}{2}\|\sqrt{\tilde{\eta}_t} \circ (\nabla F(\bar{\mathbf{w}}_{t-1}^r) - \bar{\mathbf{m}}_t^r)\|^2 - \frac{1}{2}\|\sqrt{\tilde{\eta}_t} \circ \nabla F(\bar{\mathbf{w}}_{t-1}^r)\|^2 + \frac{L}{2}\|\tilde{\eta}^2 \circ \bar{\mathbf{m}}_t^r\|^2 - \frac{1}{2}\|\sqrt{\tilde{\eta}_t} \circ \bar{\mathbf{m}}_t^r\|^2$$

$$\leq F(\bar{\mathbf{w}}_{t-1}^r) + \frac{\eta}{2\tau}\|\nabla F(\bar{\mathbf{w}}_{t-1}^r) - \bar{\mathbf{m}}_t^r\|^2 - \frac{\eta}{2(G+\tau)}\|\nabla F(\bar{\mathbf{w}}_{t-1}^r)\|^2 + \frac{\eta^2 L/\tau^2 - \eta/(G+\tau)}{2}\|\bar{\mathbf{m}}_t^r\|^2$$

$$\leq F(\bar{\mathbf{w}}_{t-1}^r) + \frac{\eta}{\tau}\|\nabla F(\bar{\mathbf{w}}_{t-1}^r) - \bar{\mathbf{m}}_{t-1}^r\|^2 + \frac{\eta}{\tau}\|\bar{\mathbf{m}}_{t-1}^r - \bar{\mathbf{m}}_t^r\|^2 - \frac{\eta}{2(G+\tau)}\|\nabla F(\bar{\mathbf{w}}_{t-1}^r)\|^2$$

$$\leq F(\bar{\mathbf{w}}_{t-1}^r) + \frac{\eta}{\tau}\|\nabla F(\bar{\mathbf{w}}_{t-1}^r) - \bar{\mathbf{m}}_{t-1}^r\|^2 + \frac{\eta}{\tau}\beta_3^2 G^2 - \frac{\eta}{2(G+\tau)}\|\nabla F(\bar{\mathbf{w}}_{t-1}^r)\|^2$$

$$(59)$$

Therefore,

$$\mathbb{E}\left[\frac{1}{RI}\sum_{r=1}^{R}\sum_{t=1}^{I}\|\nabla F(\bar{\mathbf{w}}_{t-1}^r)\|^2\right] \leq O\left(\frac{1}{\eta RI} + \beta_1 \sigma^2 + \beta_3^2 I^2 G^2 + \eta L G^2\right). \quad (60)$$

We conclude the proof by setting the parameters as the theorem.

## D  LocalAdam Algorithm

In this section, we present Algorithm 4, which uses Adam type updates in local steps to solve an ERM problem.

---

**Algorithm 4** LocalAdam

---

1: Initialization: $\bar{\mathbf{w}}^1, \bar{\mathbf{m}}^1, \bar{\mathbf{q}}^1$
2: **for** $r = 1, ..., R$ **do**
3:     $\mathbf{w}_{i,0}^r = \bar{\mathbf{w}}^r, \mathbf{m}_{i,0}^r = \bar{\mathbf{m}}^r, \mathbf{q}_{i,0}^r = \bar{\mathbf{q}}^r$
4:     **for** $t = 1, ..., I$ **do**
5:         Each machine samples data $\mathbf{z}_{i,t}^r$
6:         $\mathbf{h}_{i,t}^r = \nabla \ell(\mathbf{w}_{i,t-1}^r; \mathbf{z}_{i,t}^r)$
7:         $\mathbf{m}_{i,t}^r = (1-\beta_3)\mathbf{m}_{i,t-1}^r + \beta_3 \mathbf{h}_{i,t}^r, \mathbf{q}_{i,t}^r = (1-\beta_4)\mathbf{q}_{i,t-1}^r + \beta_4 (\mathbf{h}_{i,t}^r)^2$
8:         $\mathbf{w}_{i,t}^r = \mathbf{w}_{i,t-1}^r - \eta \frac{\mathbf{m}_{i,t}^r}{\sqrt{\mathbf{q}_{i,t}^r} + \tau}$
9:     **end for**
10:    $\bar{\mathbf{w}}^{r+1} = \frac{1}{N}\sum_{i=1}^{N}\mathbf{w}_{i,I}^r, \bar{\mathbf{m}}^{r+1} = \frac{1}{N}\sum_{i=1}^{N}\mathbf{m}_{i,I}^r$ and $\bar{\mathbf{q}}^{r+1} = \frac{1}{N}\sum_{i=1}^{N}\mathbf{q}_{i,I}^r$
11: **end for**

---

$\square$

## E  Statistics of Datasets

Table 4 summarizes the sizes of the datasets used. Table 5 summarizes the client imbalance ratio and the class imbalance ratio. The client imbalance ratio represents the ratio between the number of training samples on the client with the most data and the client with the least data, and the class imbalance ratio reflects the ratio of training data in the largest to the smallest classes in classification tasks.

## F  Running Time

Running time is reported in Tabel 6. Each algorithm was run on a high performance cluster where each machine uses a NVIDIA A100 GPU.

Table 4: Number of data for each split, with the number of training domains in the brackets. Number of training clients in the last column, with data from the same training domain to be on the same client.

|  | Train | Validation | Test | Num of Clients |
|---|---|---|---|---|
| Pile | 192,912,246 (17) | 193,105 (17) | 193,105 (17) | 17 |
| Civilcomments | 269,038 (4) | 45,180 (16) | 133,782 (16) | 4 |
| PovertyMap | 9,792 (13) | 3,936 (5) | 3,968 (5) | 13 |
| iWildsCam | 129,809 (243) | 7,314 (243) | 8,154 (243) | 8 |
| Camelyon | 302,436 (3) | 34,904 (1) | 85,054 (1) | 3 |

Table 5: Imbalance Ratio

| Datasets | Pile | CivilComments | Camelyon17 | iWildCam2020 | PovertyMap |
|---|---|---|---|---|---|
| Client Imbalance Ratio | 258 | 36.2 | 1 | 1.7 | 5.9 |
| Class Imbalance Ratio | N/A | 4.6 | 1 | 48021 | N/A |

# G    Experiments on Cifar 10

We create an imbalance dataset by reducing the data of 5 classes by 80% and then distributed the data across 100 clients according to two different Dirichlet distributions: Dirichlet (0.3) and Dirichlet (10), using code released by [61]. We use a two layer CNN as the model. Results of algorithms are summarized in Table 7. Figure 2 and Figure 2 illustrate the communication complexity of each method by comparing the worst-case testing accuracy against the number of local updates and the communicated data sizes.

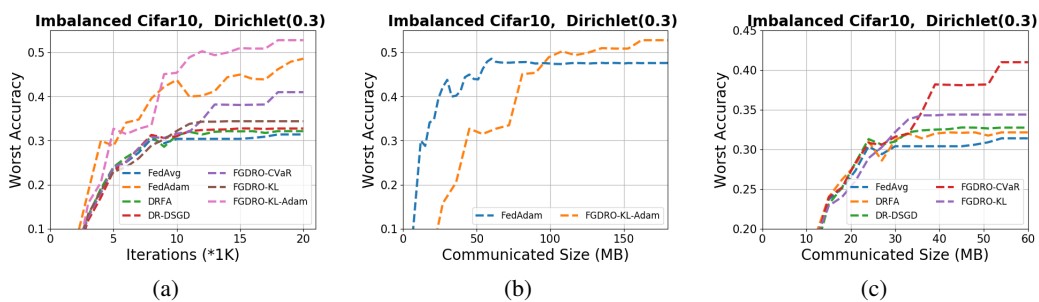

Figure 2: Convergence on Imbalanced Cifar10 with Dirichlet(0.3)

# H    On the Statistical Significance of Proposed Algorithms

In Table 8, 9 and 10, we evaluate the statistical significance of our algorithms' advantages over the baselines. In each cell, the three letters (representing FGDRO-CVaR, FGDRO-KL, and FGDRO-KL-Adam, respectively) indicate whether each of our algorithms has outperformed the corresponding baseline in that row with a confidence level greater than 95% (p-value less than 0.05). 'Y' indicates Yes, and 'N' indicates No.

Table 6: Average running time of federated algorithms. We report running (in hours) for each algorithm to finish (200K iterations for Pile data and 20K for others).

|  | Pile | CivilComments | Camelyon17 | iWildCam2020 | PovertyMap |
|---|---|---|---|---|---|
| FedAvg | 29.77 | 3.05 | 4.59 | 12.90 | 1.77 |
| FedAdam | 35.39 | 3.13 | 6.01 | 12.91 | 3.22 |
| DRFA | 34.12 | 3.27 | 4.94 | 12.95 | 2.25 |
| DR-DSGD | 25.26 | 4.01 | 6.26 | 13.02 | 4.71 |
| FGDRO-CVAR | 34.07 | 4.03 | 7.19 | 13.01 | 5.02 |
| FGDRO-KL | 34.92 | 3.65 | 7.23 | 14.57 | 5.42 |
| FGDRO-KL-Adam | 35.98 | 3.80 | 7.54 | 14.82 | 5.50 |

Table 7: Imbalanced Cifar10, data alloacted to clients accoring to Dirichlet distributions

| Datasets | Cifar10, Dirichlet(0.3) | | Cifar10, Dirichlet(10) | |
|---|---|---|---|---|
| Metric | Worst Acc | Average Acc | Worst Acc | Average Acc |
| FedAvg | 0.3140 ($\pm$0.0027) | 0.6236 ($\pm$0.0019) | 0.3620 ($\pm$0.0032) | 0.6742 ($\pm$0.0020) |
| SCAFFOLD | 0.3245 ($\pm$0.0032) | 0.6337 ($\pm$0.0038) | 0.3821 ($\pm$0.0022) | 0.6816 ($\pm$0.0025) |
| FedProx | 0.3102 ($\pm$0.0027) | 0.6189 ($\pm$0.0036) | 0.3757 ($\pm$0.0041) | 0.6925 ($\pm$0.0053) |
| FedAdam | 0.4860 ($\pm$0.0047) | **0.7147** ($\pm$0.0058) | 0.4460 ($\pm$0.0039) | 0.7042 ($\pm$0.0043) |
| DRFA | 0.3215 ($\pm$0.0129) | 0.6381 ($\pm$0.0131) | 0.3752 ($\pm$0.0053) | 0.6739 ($\pm$0.0048) |
| DR-DSGD | 0.3277 ($\pm$0.0074) | 0.6403 ($\pm$0.0058) | 0.3700 ($\pm$0.0063) | 0.6792 ($\pm$0.0088) |
| FGDRO-CVaR | 0.4100 ($\pm$0.0032) | 0.6606 ($\pm$0.0039) | 0.4010 ($\pm$0.0022) | 0.6882 ($\pm$0.0027) |
| FGDRO-KL | 0.3560 ($\pm$0.0018) | 0.6369 ($\pm$0.0029) | 0.4110 ($\pm$0.0035) | 0.6951 ($\pm$0.0042) |
| FGDRO-KL-Adam | **0.5280** ($\pm$0.0101) | 0.7057 ($\pm$0.0133) | **0.5110** ($\pm$0.0072) | **0.7286** ($\pm$0.0063) |

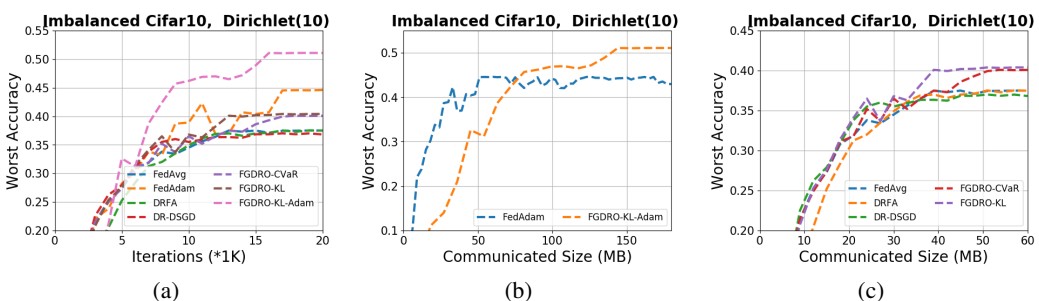

(a)  (b)  (c)

Figure 3: Convergence on Imbalanced Cifar10 with Dirichlet(10)

Table 8: Statistical Confidence on Imbalanced Cifar10

| Datasets | Cifar10, Dirichlet(0.3) | | Cifar10, Dirichlet(10) | |
|---|---|---|---|---|
| Metric | Worst Acc | Average Acc | Worst Acc | Average Acc |
| FedAvg | Y\|Y\|Y | Y\|Y\|Y | Y\|Y\|Y | Y\|Y\|Y |
| SCAFFOLD | Y\|Y\|Y | Y\|N\|Y | Y\|Y\|Y | Y\|Y\|Y |
| FedProx | Y\|Y\|Y | Y\|Y\|Y | Y\|Y\|Y | N\|N\|Y |
| FedAdam | N\|N\|Y | N\|N\|N | N\|N\|Y | N\|N\|Y |
| DRFA | Y\|Y\|Y | Y\|N\|Y | Y\|Y\|Y | Y\|Y\|Y |
| DR-DSGD | Y\|Y\|Y | Y\|N\|Y | Y\|Y\|Y | Y\|Y\|Y |

Table 9: Statistical Confidence on Piles and CivilComments

| Datasets | Pile | | CivilComments | |
|---|---|---|---|---|
| Metric | Worst Log-PPL | Average Log-PPL | Worst Acc | Average Acc |
| FedAvg | N\|Y\|Y | N\|Y\|Y | Y\|Y\|Y | N\|Y\|N |
| SCAFFOLD | N\|Y\|Y | N\|Y\|Y | Y\|Y\|Y | N\|Y\|N |
| FedProx | N\|Y\|Y | N\|Y\|Y | Y\|Y\|Y | N\|Y\|Y |
| FedAdam | N\|N\|Y | N\|N\|Y | Y\|Y\|Y | N\|Y\|N |
| DRFA | N\|Y\|Y | N\|Y\|Y | Y\|Y\|Y | Y\|Y\|Y |
| DR-DSGD | N\|Y\|Y | N\|Y\|Y | Y\|Y\|Y | Y\|Y\|Y |

Table 10: Statistical Confidence on Camelyon17, iWildCam2020, and PovertyMap

| Datasets | Camelyon17 | iWildCam2020 | CivilComments | |
|---|---|---|---|---|
| Metric | Acc | Macro F1 | Worst Pearson | Average Pearson |
| FedAvg | N\|Y\|Y | Y\|Y\|N | Y\|N\|Y | Y\|Y\|Y |
| SCAFFOLD | N\|Y\|Y | Y\|Y\|N | Y\|N\|Y | Y\|N\|Y |
| FedProx | N\|Y\|Y | Y\|Y\|Y | Y\|N\|Y | Y\|Y\|Y |
| FedAdam | N\|N\|N | Y\|Y\|Y | Y\|N\|Y | N\|N\|N |
| DRFA | Y\|Y\|Y | Y\|Y\|Y | Y\|Y\|Y | Y\|Y\|Y |
| DR-DSGD | Y\|Y\|Y | Y\|Y\|Y | Y\|Y\|Y | Y\|Y\|Y |

