# OpenReview forum: "Communication-Efficient Federated Group Distributionally Robust Optimization"
_NeurIPS.cc/2024/Conference — NeurIPS 2024 poster_

### Official Review · Reviewer_tWjT · 2024-06-20

**Soundness:** 3
**Presentation:** 2
**Contribution:** 3
**Rating:** 5
**Confidence:** 3

**Summary:**

This work introduces three algorithms for communication-efficient Federated Group Distributionally Robust Optimization. The effectiveness of the proposed algorithms are verified through both theoretical and experimental results.

**Strengths:**

1) This work studies an important problem of federated group distributionally robust optimization.
2) The theoretical results show the advantages of the proposed algorithms.

**Weaknesses:**

1) This work proposes three algorithms, including FGDRO-CVaR, FGDRO-KL, and FGDRO-KL-Adam. There lacks a comparison between these algorithms. For example, what are the connections and differences between these algorithms?
2) The analysis for FGDRO-CVaR assumes the loss function to be rho-weakly convex, which is missing from the main context.

**Questions:**

1) Missing reference: How about the comparison with this work [1]?

[1] Communication-Efficient Distributionally Robust Decentralized Learning https://arxiv.org/pdf/2205.15614

2) What are the experimental setups for the number of clients and non-IID?

3) In experimental results (Tables 2 and 3), FGDRO-CVaR seems to have no advantages in both task; why do we need this algorithm? Besides, it is better to highlight the best-performance results in Tables 2 and 3.

4) Intuitively, using Adam optimizer can bring training speedup and is supposed to outperform other algorithms. But why do the results show that sometimes FGDRO-KL is better than FGDRO-KL-Adam?

5) In proof, is an assumption of bounded gradient needed? If I don't misunderstand, Line 550 indicates such an assumption.

6) If the loss function assumes to be convex, analyzing the optimal distance between the loss value the minimum loss should be better in Theorem 6.2.

**Limitations:**

See weakness.

---

> ### Author Rebuttal · Authors · 2024-08-07
>
> Thank you for the review! We believe we can address your concerns as follows.
>
> ***Q1: There lacks a comparison between the three proposed algorithms. What are the connections and differences between these algorithms?***
>
> ***A:*** FGDRO-CVaR and FGDRO-KL employ well-established regularization techniques [Deng et al., 2020; Lan & Zhang, 2023], each suited to different tasks and data distributions.
>
> __FGDRO-CVaR__ focuses on optimizing for worst-case scenarios or the average of the worst-case losses, making it particularly effective in high-stakes applications like healthcare and finance, where avoiding extreme losses is crucial. However, it can be sensitive to outliers or malicious client attacks.
> __FGDRO-KL__, on the other hand, uses KL divergence as a softer regularizer to promote smoother and more stable learning. Thus it can be beneficial in scenarios where robustness to outliers or malicious clients is needed.
> __FGDRO-KL-Adam__ further enhances FGDRO-KL by incorporating Adam-type updates.
>
> [Deng et. al., 2020] Distributionally Robust Federated Averaging.
>
> [Lan & Zhang, 2023] Optimal Methods for Convex Risk Averse Distributed Optimization
>
> ***Q2: The analysis for FGDRO assumes the loss function to be $\rho$-weakly convex, which is missing from the main context.***
>
> ***A:*** This property can indeed be deduced from Assumption 4.1, as demonstrated in equation (16) of the appendix. In the revised version, we will ensure that this assumption is clearly articulated within the main body.
>
> ***Q3: Missing reference "Communication-Efficient Distributionally Robust Decentralized Learning"***
>
> ***A:*** Thank you for bringing this literature to our attention. While this literature addresses similar problem to ours, it is tailored for decentralized federated learning. It would incur high communication cost if directly applied to a centralized federated learning framework because it requires communication at every iteration. Additionally, the approach discussed is not applicable to scenarios involving a CVaR regularizer. We will include a discussion of this literature in our revision.
>
> ***Q4: What are the experimental setups for the number of clients and non-IID?***
>
> ***A:***
> In our previously submitted experiments, we used a natural data split, where data from the same hospital, web source, location, or demographic group were assigned to the same machine, with up to 17 clients in total. The following table summarizes key statistics of the data, including the Client Imbalance Ratio, which represents the ratio between the number of training samples on the client with the most data and the client with the least data, and the Class Imbalance Ratio, which reflects the ratio of training data in the largest to the smallest classes in classification tasks.
> |Datasets |__Pile__ |  __CivilComments__ |  __Camelyon17__ | __iWildCam2020__ |  __PovertyMap__ |
> |------|-------|-------|------|----|-------|
> |Client Imbalance Ratio | 258 | 36.2 | 1 | 1.7 |  5.9  |
> |Class Imbalance Ratio | N/A | 4.6 | 1 | 48021 | N/A |
>
> To further explore non-IID scenarios, we conducted additional experiments using the Cifar10 dataset, where we created an imbalance by reducing the representation of 5 classes by 80%, and then distributed the data across 100 clients using two different Dirichlet distributions: Dirichlet (0.3) and Dirichlet (10). Results are summarized as below (detailed table at Section 2 of https://anonymous.4open.science/r/NeurIPS_11919-3402).
>
> |Datasets |  Dirichlet(0.3) |  Dirichlet(10) |
> |--|--|-|
> |Metric  | Worst Acc, Average Acc |  Worst Acc, Average Acc |
> |FedAvg |0.3140, 0.6236 | 0.3620, 0.6742 |
> |SCAFFOLD|0.3245, 0.6337| 0.3821, 0.6816|
> |FedProx | 0.3102, 0.6189 | 0.3757, 0.6925|
> |FedAdam| 0.4860, __0.7147__ | 0.4460, 0.7042|
> |DRFA | 0.3215, 0.6381 | 0.3752, 0.6739 |
> |DR-DSGD | 0.3277, 0.6403 | 0.3700, 0.6792 |
> |FGDRO-CVaR | 0.4100, 0.6606 | 0.4010, 0.6882|
> |FGDRO-KL| 0.3560, 0.6369| 0.4110, 0.6951|
> |FGDRO-KL-Adam | __0.5280__, 0.7057 | __0.5110__, __0.7286__|
>
> ***Q5: FGDRO-CVaR seems to have no advantages in both tasks; why do we need this algorithm? Besides, it is better to highlight the best-performance results in Tables 2 and 3.***
>
> ***A:***
> On the CivilComments, iWildCam2020, PovertyMap and newly added Cifar10 datasets, FGDRO-CVaR outperforms previous baselines. Specifically, on PovertyMap, it surpasses FGDRO-KL, although it performs worse than FGDRO-KL-Adam. The primary reason for this is that FGDRO-CVaR does not implement Adam-type local updates. This is due to the current limitations in developing provable algorithms for FGDRO-CVaR with Adam-type updates, given the nonsmoothness of the problem. This is an area we plan to explore in future research. We will also highlight the best-performing results in the tables in our revision. Thank you for the suggestion.
>
> ***Q6: Why do the results show that sometimes FGDRO-KL is better than FGDRO-KL-Adam?***
> ***A:*** It is not unusual for SGD updates to occasionally outperform Adam in specific scenarios. Although Adam is generally more robust, it comes with additional hyperparameters that require careful tuning. We believe the observed result where FGDRO-KL-Adam underperforms compared to FGDRO-KL could be improved through an extensive hyperparameter search.
>
> ***Q7: In proof, is an assumption of bounded gradient needed?***
>
> ***A:*** Yes, an assumption of bounded gradients is necessary. This requirement is implicitly addressed in Assumptions 4.1 and 5.1 of the main text, where we assume that the functions are Lipschitz continuous. That implies the bounded gradients. We will explicitly address this point in the main text of our upcoming revision.
>
> ***Q8: If the loss function assumes to be convex, analyzing the optimal distance between the loss value the minimum loss should be better in Theorem 6.2.***
>
> ***A:*** While this is a valuable suggestion, our work does not assume convexity in the loss function, as the focus of this paper is on the nonconvex regime.

---

> > ### Comment · Reviewer_tWjT · 2024-08-08
> > **Thank you for the response**
> >
> > After reading other reviews and the authors' responses, I would keep my score. My major concern is that the bounded gradient assumption is somehow too strong.

---

> > > ### Author Response · Authors · 2024-08-10
> > > **Regarding your concern about the bounded stochastic gradient assumption**
> > >
> > > Thank you for your valuable feedback! After a thorough analysis, we have managed to relax the bounded stochastic gradient assumption ($||\nabla \ell(\mathbf{w};\mathbf{z})||^2\leq G^2$) to more standard assumptions in the literature of federated learning and optimization. Specifically, we now assume a bounded true gradient ($||\nabla \ell(\mathbf{w};\mathcal{D}\_i)||^2\leq C_\ell^2$, where $\ell(\mathbf{w};\mathcal{D}\_i) = \mathbb{E}\_{\mathbf{z}\in\mathcal{D}\_i} \ell(\mathbf{w}; \mathcal{D}\_i)$) and bounded variance ($\mathbb{E}_{\mathbf{z} \in \mathcal{D}\_i}||\ell(\mathbf{w}; \mathbf{z}) - \ell(\mathbf{w}; \mathcal{D}\_i)||^2 \leq \sigma^2$). We have updated the assumption and analysis accordingly, and you can find the revised details in the "updated_analysis.pdf" file at https://anonymous.4open.science/r/NeurIPS_11919-3402. For example, your original concern about the usage of the bounded stochastic gradient assumption in Line 550 has now been addressed with these relaxed assumptions in Lines 550-551.

---

### Official Review · Reviewer_f6MF · 2024-07-08

**Soundness:** 3
**Presentation:** 3
**Contribution:** 3
**Rating:** 6
**Confidence:** 3

**Summary:**

This paper addresses the challenge of reducing communication costs and sample complexity in Federated Group Distributionally Robust Optimization (FGDRO). The authors present the FGDRO-CVaR algorithm and the FGDRO-KL algorithm to address different constraints. Subsequently, they conduct extensive experiments across various real-world tasks, including NLP and CV tasks. The corresponding empirical results confirm the effectiveness of their proposed methods.

**Strengths:**

1. The exploration of reducing communication costs for federated group DRO is a rarely-studied topic within the FL community.

2. The theoretical convergence analysis for the proposed algorithms is somewhat solid.

3. The authors conduct comprehensive experiments to validate the effectiveness of the devised algorithms.

**Weaknesses:**

1. The contributions and novelties of this paper are unclear. It appears that the authors have directly combined existing federated adaptive algorithms with pre-existing federated group DRO methods in this paper.

**Questions:**

1. The introduction's treatment of the concept of generalization appears incomplete. It is evident that there are two levels of generalization in Federated Learning, as delineated in [1] and [2].

2. As highlighted in the aforementioned weaknesses, the authors should provide additional clarification regarding the contributions and novelty of this paper. Overall, it appears that the proposed method is primarily a direct combination of existing methods.

3. Similarly, the authors should delineate the challenges and innovations intrinsic to their theoretical analysis. Specifically, they should underscore the complexities involved in analyzing federated adaptive algorithms when applied in federated group DRO.

4. On line 154 of Page 4, what is the relationship between the "accurate estimate" and the "moving average" in the subsequent sentence?

5. Some minor points to address. The authors might consider offering more empirical results on convergence analysis in the experimental section. Additionally, they should further consider the statistical significance of these convergence analyses.

[1] Hu X, Li S, Liu Y. Generalization bounds for federated learning: Fast rates, unparticipating clients and unbounded losses[C]//The Eleventh International Conference on Learning Representations. 2023.

[2] Yuan H, Morningstar W, Ning L, et al. What do we mean by generalization in federated learning?[J]. arxiv preprint arxiv:2110.14216, 2021.

**Limitations:**

The authors have adequately addressed the limitations and potential negative societal impact of their work.

---

> ### Author Rebuttal · Authors · 2024-08-07
>
> Thank you for the review! We address your suggestions and concerns as follows.
>
> ***Q1: The introduction's treatment of the concept of generalization appears incomplete. It is evident that there are two levels of generalization in Federated Learning, as delineated in two literature.***
>
> ***A:***
> Thank you for bringing to our attention the literature that discusses the two levels of generalization in federated learning. In our work, we are referring to generalization at the participant level, where the goal is to develop robust models that perform well across both participating and non-participating clients. Based on the insights from the mentioned literature, we have revised the first paragraph of our introduction as follows:
>
> _"... Generalization here refers to the model's ability to perform consistently across different clients, including those that have not participated in the training [Hu et al., 2023, Yuan et al., 2021]."_
>
> ***Q2: It appears that the authors have directly combined existing federated adaptive algorithms with pre-existing federated group DRO methods in this paper.***
>
> ***A:***
> First, our federated adaptive algorithms differ from those in the existing literature in terms of both algorithm design and analysis, such as [Reddi et al., 2020], as we employ Adam-type updates at every step while  [Reddi et al., 2020] uses SGD updates on local steps and only apply Adam at global steps. Second, our approach to federated group DRO is distinct from previous works. By considering constraint-free equivalent forms of federated group DRO problems and focusing on dedicated algorithm design and analysis, we have achieved lower communication and sample complexity, with or without adaptive components, compared to the literature.
>
> ***Q3: The authors should delineate the challenges and innovations intrinsic to their theoretical analysis.***
>
> ***A:***
> For FGDRO-CVaR, we are the first to consider a constraint-free equivalent form and develop a communication-efficient algorithm for it, significantly reducing communication costs, as evidenced in Table 1. In addition to sharing machine learning models, we propose introducing only an additional scalar threshold to select participants in each round, minimizing additional costs. The new formulation, being a nonsmooth and compositional problem, introduces challenges that are uncommon in federated learning literature. We addressed these by carefully designing the use of moving average estimators to handle client drift and achieve linear speedup.
>
> For FGDRO-KL, although previous literature has considered constraint-free compositional reformulations, they typically require large batch sizes on each machine to estimate gradients. This approach is impractical and incurs high sample complexity. Instead, we employ moving averages that only require small data batches while still providing accurate gradient estimations, making our method more efficient.
>
> For FGDRO-KL-Adam, we allow local updates using Adam-type updates, which introduces the challenge of handling unbiased gradients, further complicated by the use and updating of the second-order moment. Our analysis carefully manages the moving estimates of the first and second-order moments, ensuring that the solution provably converges.
>
> ***Q4: On line 154 of Page 4, what is the relationship between the "accurate estimate" and the "moving average" in the subsequent sentence?***
>
> ***A:*** The "moving average" is what we refer to as our "accurate estimate." To clarify this relationship, we have restated the sentences as follows: _"To address this, it is common practice to create an accurate estimate of
> $g_i(w)$ [23, 22, 60, 61, 31]. Specifically, we employ a moving average $u$ as our accurate estimate."_
>
> ***Q5: Some minor points to address. The authors might consider offering more empirical results on convergence analysis in the experimental section. Additionally, they should further consider the statistical significance of these convergence analyses.***
>
> ***A:*** We conducted additional experiments using the Cifar10 dataset, where we created an imbalance dataset by reducing the data of 5 classes by 80%, and then distributed the data across 100 clients using two different Dirichlet distributions: Dirichlet (0.3) and Dirichlet (10). Results are summarized as below (detailed table at Section 2 of https://anonymous.4open.science/r/NeurIPS_11919-3402).
>
> |Datasets |  Dirichlet(0.3) |  Dirichlet(10) |
> |-----------------|-----------------|-----------------|
> |Metric  | Worst Acc, Average Acc |  Worst Acc, Average Acc |
> |FedAvg |0.3140, 0.6236 | 0.3620, 0.6742 |
> |SCAFFOLD|0.3245, 0.6337| 0.3821, 0.6816|
> |FedProx | 0.3102, 0.6189 | 0.3757, 0.6925|
> |FedAdam| 0.4860, __0.7147__ | 0.4460, 0.7042|
> |DRFA | 0.3215, 0.6381 | 0.3752, 0.6739 |
> |DR-DSGD | 0.3277, 0.6403 | 0.3700, 0.6792 |
> |FGDRO-CVaR | 0.4100, 0.6606 | 0.4010, 0.6882|
> |FGDRO-KL| 0.3560, 0.6369| 0.4110, 0.6951|
> |FGDRO-KL-Adam | __0.5280__, 0.7057 | __0.5110__, __0.7286__|
>
> We have expanded our analysis to better address communication efficiency. Specifically, we now include figures that illustrate the communication complexity of each method by comparing the worst-case testing accuracy against the number of local updates and the communicated data sizes. You can find the figures at Section 1 of https://anonymous.4open.science/r/NeurIPS_11919-3402.

---

> > ### Comment · Reviewer_f6MF · 2024-08-08
> >
> > Thanks to the authors for the additional explanations of the other issues that I have mentioned in the review and more experiments, which have solved all of my concerns. I recommend the acceptance of this paper. However, since I am not an expert in this field, I decide to keep my current score.

---

### Official Review · Reviewer_sEsv · 2024-07-12

**Soundness:** 3
**Presentation:** 4
**Contribution:** 2
**Rating:** 4
**Confidence:** 5

**Summary:**

The paper presents three methods for Federated Learning Group Distributionally Robust Optimization: (i) one tailored to reduce the CVaR which optimizes the top K-losses, (ii) another one tailored to tackle the KL divergence, and finally (iii) one that uses Adam locally. The paper is well written and the ideas are presented. To the best of my knowledge, the proofs are correct. My main concerns are regarding the relevance and importance of the subject, the lack of experiments, and the lack of empirical studies on communication efficiency.

**Strengths:**

[S1] The paper is well-written, and the ideas are presented.

[S2] The theoretical results are correct, to the best of my knowledge.

**Weaknesses:**

[W1] The relevance of the subject is not entirely addressed. See [Q1]

[W2] The experiment section is limited. In particular, the paper does not present any intuition on the problems they are solving. They do not consider the number of samples per server for example. I believe the authors should include a class imbalance problem [AN AGNOSTIC APPROACH TO FEDERATED LEARNING WITH CLASS IMBALANCE - Shen et al, ICLR 22].

[W3] Communication efficiency is not properly addressed by the authors. The authors show the number of communication rounds required, but they do not take into account how much is communicated. The authors claim that this method is more efficient in terms of communication, and they show it theoretically, but in the experiment section, there is no evidence of communication efficiency. I suggest the authors reveal the communication cost associated with each method, measured in the amount of data shared between servers.

[W4] Privacy is an important subject of Federated Learning, but in this paper, there is no analysis of the privacy aspect. Can the authors elaborate on the privacy aspect of this work?

[W5] Federated learning is a technique used to train on a set of machines. The idea is that the number of machines that participate is large. It appears to me that the largest number of servers is 17. This seems to me insufficient for a distributed learning problem.

**Questions:**

[Q1] Why should be designed solutions that are distributional robust? And, at what cost? If we compare a method that simply maximizes/minimizes the FL problem, what is the overall loss? I believe the overall loss should be smaller, given that being distributionally robust is a particular case, and therefore, the unconstrained problem achieves a smaller minimum.

**Limitations:**

See weaknesses.

---

> ### Author Rebuttal · Authors · 2024-08-07
>
> Thank you for your time to review! Below we address your concerns and suggestions.
>
> ***Q1: Why should be designed solutions that are distributional robust? And, at what cost? If we compare a method that simply maximizes/minimizes the FL problem, what is the overall loss?***
>
> ***A:*** Designing distributionally robust solutions in federated learning is essential for two reasons:  __1)Robustness__: It effectively manages distributional shifts, making models more reliable in real-world scenarios. In high-stakes applications like healthcare and finance, where failure in extreme cases can be costly, distributional robustness is crucial for minimizing risks. __2) Fairness__: By upweighting the group that has lower performance, it aims to not only perform good on average but on every subpopulations equitably.
>
> Focusing on distributional robustness may result in a potential reduction in average performance for the majority in the observed distribution. However, this trade-off is necessary to achieve the broader goal of creating models that are both robust to distributional shifts during testing and fair across all data groups.
>
> ***Q2: In particular, the paper does not present any intuition on the problems they are solving. They do not consider the number of samples per server for example. I believe the authors should include a class imbalance problem [AN AGNOSTIC APPROACH TO FEDERATED LEARNING WITH CLASS IMBALANCE - Shen et al, ICLR 22].***
>
> ***A:*** In our previous submitted experiments, we utilized natural data splits where data from the same hospital, web source, locations, or demographic groups were placed on the same machine. These experiments have involved with highly imbalanced number of data on servers. The following table summarizes key statistics of the data, including the Client Imbalance Ratio, which represents the ratio between the number of training samples on the client with the most data and the client with the least data, and the Class Imbalance Ratio, which reflects the ratio of training data in the largest to the smallest classes in classification tasks.
>
> |Datasets |__Pile__ |  __CivilComments__ |  __Camelyon17__ | __iWildCam2020__ |  __PovertyMap__ |
> |------|-------|-------|------|----|-------|
> |Client Imbalance Ratio | 258 | 36.2 | 1 | 1.7 |  5.9  |
> |Class Imbalance Ratio | N/A | 4.6 | 1 | 48021 | N/A |
>
> To further address this concern, we have conducted additional experiments using the Cifar10 dataset. We create an imbalance dataset by reducing the data of 5 classes by 80\% and then distributed the data across 100 clients according to two different Dirichlet distributions: Dirichlet (0.3) and Dirichlet (10), using code released by [Shen et al, ICLR 22]. Results are summarized as below (detailed table at Section 2 of https://anonymous.4open.science/r/NeurIPS_11919-3402).
>
> |Datasets |  Dirichlet(0.3) |  Dirichlet(10) |
> |-----------------|-----------------|-----------------|
> |Metric  | Worst Acc, Average Acc |  Worst Acc, Average Acc |
> |FedAvg |0.3140, 0.6236 | 0.3620, 0.6742 |
> |SCAFFOLD|0.3245, 0.6337| 0.3821, 0.6816|
> |FedProx | 0.3102, 0.6189 | 0.3757, 0.6925|
> |FedAdam| 0.4860, __0.7147__ | 0.4460, 0.7042|
> |DRFA | 0.3215, 0.6381 | 0.3752, 0.6739 |
> |DR-DSGD | 0.3277, 0.6403 | 0.3700, 0.6792 |
> |FGDRO-CVaR | 0.4100, 0.6606 | 0.4010, 0.6882|
> |FGDRO-KL| 0.3560, 0.6369| 0.4110, 0.6951|
> |FGDRO-KL-Adam | __0.5280__, 0.7057 | __0.5110__, __0.7286__|
>
> ***Q3: Communication efficiency is not properly addressed by the authors. I suggest the authors reveal the communication cost associated with each method, measured in the amount of data shared between servers.***
>
> ***A:*** We have expanded our analysis to better address communication efficiency. Specifically, we now include figures that illustrate the communication complexity of each method by comparing the worst-case testing accuracy against the number of local updates and the communicated data sizes. You can find the figures at Section 1 of https://anonymous.4open.science/r/NeurIPS_11919-3402.
>
> ***Q4: Privacy is an important subject of Federated Learning, but in this paper, there is no analysis of the privacy aspect. Can the authors elaborate on the privacy aspect of this work?***
>
> ***A:***
> Thank you for your comment. In our work, models and estimates of gradients are shared, both of which have been extensively studied in the privacy literature. The techniques developed in these studies can be integrated with our algorithms and applied within our framework.
>
> Additionally, we aggregate certain scalar variables (in FGDRO-CVaR, the scalar variable s; in FGDRO-KL and FGDRO-KL-Adam, the scalar variable v). Borrowing the technique from the Remark 3.1 of [Shen et al., ICLR 2022], these variables can be aggregated using Homomorphic Encryption, ensuring that their exact values remain confidential. We appreciate the reviewer for bringing this literature to our attention.
>
> In summary, while we acknowledge the importance of privacy in federated learning, we believe that our algorithms do not introduce significant new challenges in this area. We will included an elaborated discussion on this matter in the revision.
>
> ***Q5: Federated learning is a technique used to train on a set of machines. The idea is that the number of machines that participate is large. It appears to me that the largest number of servers is 17.***
>
> ***A:*** In our new experiments on Cifar10, we have scaled up the number of clients to 100. Please check the results presented in Section 1 and 2 of https://anonymous.4open.science/r/NeurIPS_11919-3402.

---

> > ### Comment · Reviewer_sEsv · 2024-08-09
> > **Reponse**
> >
> > I appreciate the author's response. I choose to keep my score.

---

### Official Review · Reviewer_rzqA · 2024-07-13

**Soundness:** 3
**Presentation:** 2
**Contribution:** 3
**Rating:** 5
**Confidence:** 2

**Summary:**

This paper aims to improve the efficiency of existing federated group distributionally robust optimization (FGDRO) when considering two specific types of regularization, condition value at risk and KL divergence. To address the first type of problem, the authors propose FGDRO-CVaR that reduces the sample complexity and communication costs simultaneously. For KL conditions, the proposed FGDRO-KL reduces the sample complexity while retaining the same communication costs. Moreover, the authors integrate the notion of Adam into FGDRO-KL, yielding FGDRO-KL-Adam and achieving better convergence speed.

**Strengths:**

1. The paper is well-written, though some background information is missing.
2. The problem is well-motivated. The sample and communication efficiency is a pivotal problem in federated learning, though the benefits are not fully analyzed in the experiments.
3. The proposed method is grounded and improves over prior baselines.

**Weaknesses:**

1. The background can be more thoroughly explained. The authors are encouraged to provide additional context to address the following questions, which will greatly enhance the paper's completeness. Why is federated group distributionally robust optimization (FGDRO) an important problem or technique? What are the sources of the additional communication costs? Why is it necessary to consider two different types of regularization? Are these types of regularization relevant to different applications?

2. My major concerns lie in the experiments and their settings.

    - **Data Splits.** While FGDRO's main advantage appears to be its ability to address non-IID optimization, the experimental setup concerning data splits lacks clarity. An analysis of the non-IID levels, such as those derived from different Dirichlet-distributed data splits with varying $\lambda$ values, is missing. Including more representative baselines, such as SCAFFOLD and FedProx, which are also designed for non-IID optimization, could further enhance the analyses.

    - **Performance.** The proposed method performs similarly to the baselines in most experiments. For example, in Tables 2 and 3, apart from the Adam variant, the proposed method is comparable to the baselines. This would be acceptable if the proposed method demonstrated improved efficiency; however, relevant analyses on this aspect are absent from the experiments.

    - **Communication or Sample Complexity Analysis.** An empirical analysis comparing complexity versus utility would be beneficial and highlight the advantages of the proposed method. For instance, the experiment in Figure 1 can be extended to a comparison among different baselines.

**Questions:**

1. The datasets considered in this paper do not seem common in the existing literature. Could the authors report numbers on datasets like CIFAR-10/100 or EMNIST? Why did the authors choose the datasets in the paper?

**Limitations:**

Yes, the authors have discussed the limitations in the paper.

---

> ### Author Rebuttal · Authors · 2024-08-07
>
> Thank you for the review! We address your questions as below and will include the discussion in the revision.
>
> ***Q1: Why is FGDRO an important problem or technique?***
>
> ***A:*** In federated learning, data is distributed across multiple clients, each with its own unique data distribution. FGDRO is crucial because it addresses the heterogeneity of these distributions, ensuring that models are fair across different clients and also robust to distributional shifts.
>
> ***Q2: What are the sources of the additional communication costs?***
>
> ***A:***
> For FGDRO-CVaR, the only additional communication involves a scalar variable $s$, which is negligible compared to the large size of the deep learning model that needs to be shared in federated learning algorithms. For FGDRO-KL, the primary source of additional communication cost is the sharing of $m$, a moving average estimator of the gradient, which is the same size as the model itself. For FGDRO-KL-Adam, the additional communication cost arises from sharing both $m$ and $q$, which are estimators of the first and second order moments of gradients, respectively.
>
> ***Q3: Why is it necessary to consider two different types of regularization?***
>
> ***A:***
> FGDRO-CVaR and FGDRO-KL employ well-established regularization techniques [Deng et al., 2020; Lan & Zhang, 2023], each suited to different tasks and data distributions. __FGDRO-CVaR__ focuses on optimizing for worst-case scenarios or the average of the worst-case losses, making it particularly effective in high-stakes applications like healthcare and finance, where avoiding extreme losses is crucial. However, it can be sensitive to outliers or malicious client attacks.
> __FGDRO-KL__, on the other hand, uses KL divergence as a softer regularizer to promote smoother and more stable learning. Thus, it can be beneficial in scenarios where robustness to outliers or malicious clients is needed.
> __FGDRO-KL-Adam__ further enhances FGDRO-KL by incorporating Adam-type updates.
>
> [Deng et. al., 2020] Distributionally Robust Federated Averaging.
>
> [Lan & Zhang, 2023] Optimal Methods for Convex Risk Averse Distributed Optimization
>
> ***Q4: Data Splits. The experimental setup concerning data splits lacks clarity. An analysis of the non-IID levels, such as those derived from different Dirichlet-distributed data splits is missing.***
>
> ***A:*** In our previously submitted experiments, we used a natural data split, where data from the same hospital, web source, location, or demographic group were assigned to the same machine, with up to 17 clients in total. The following table summarizes key statistics of the data, including the Client Imbalance Ratio, which represents the ratio between the number of training samples on the client with the most data and the client with the least data, and the Class Imbalance Ratio, which reflects the ratio of training data in the largest to the smallest classes in classification tasks.
> |Datasets |__Pile__| __CivilComments__| __Camelyon17__|__iWildCam2020__| __PovertyMap__|
> |-|-|-|-|-|-|
> |Client Imbalance Ratio | 258 | 36.2 | 1 | 1.7 |  5.9  |
> |Class Imbalance Ratio | N/A | 4.6 | 1 | 48021 | N/A |
>
> We conducted additional experiments using the Cifar10 dataset, where we created an imbalance dataset by reducing the data of 5 classes by 80%, and then distributed the data across 100 clients using two different Dirichlet distributions: Dirichlet (0.3) and Dirichlet (10). Results are summarized as below (detailed table at Section 2 of https://anonymous.4open.science/r/NeurIPS_11919-3402).
>
> |Datasets |  Dirichlet(0.3) |  Dirichlet(10) |
> |-|-|-|
> |Metric  | Worst Acc, Average Acc |  Worst Acc, Average Acc |
> |FedAvg |0.3140, 0.6236 | 0.3620, 0.6742 |
> |SCAFFOLD|0.3245, 0.6337| 0.3821, 0.6816|
> |FedProx | 0.3102, 0.6189 | 0.3757, 0.6925|
> |FedAdam| 0.4860, __0.7147__ | 0.4460, 0.7042|
> |DRFA | 0.3215, 0.6381 | 0.3752, 0.6739 |
> |DR-DSGD | 0.3277, 0.6403 | 0.3700, 0.6792 |
> |FGDRO-CVaR | 0.4100, 0.6606 | 0.4010, 0.6882|
> |FGDRO-KL| 0.3560, 0.6369| 0.4110, 0.6951|
> |FGDRO-KL-Adam | __0.5280__, 0.7057 | __0.5110__, __0.7286__|
>
> ***Q5: Including more baselines, such as SCAFFOLD and FedProx, which are also designed for non-IID optimization, could further enhance the analyses.***
>
> ***A:***
> We have included SCAFFOLD and FedProx as baselines (see Section 2 and 4 of https://anonymous.4open.science/r/NeurIPS_11919-3402). However, it's important to highlight some fundamental differences between these methods and our FGDRO algorithms. SCAFFOLD and FedProx optimize the average loss, whereas our FGDRO algorithms focus on prioritizing clients with poorer performance, thereby enhancing robustness and generalization.
>
> ***Q6: The proposed method performs similarly to the baselines in most experiments. This would be acceptable if the proposed method demonstrated improved efficiency. An empirical analysis comparing complexity versus utility would be beneficial.***
>
> ***A:***
> We would like to highlight that our algorithms statistically significantly outperform the baselines in most tasks, as shown by the p-values in Section 4 of https://anonymous.4open.science/r/NeurIPS_11919-3402.
>
> We include additional figures that illustrate the complexity comparison of methods. Specifically, we compare the worst-case testing accuracy against the number of local iterations, and the worst-case testing accuracy against the size of communicated data. These figures can be found at Section 1 of https://anonymous.4open.science/r/NeurIPS_11919-3402.
>
> ***Q7: The datasets considered in this paper do not seem common in the existing literature.***
>
> ***A:***
> The datasets we used were chosen because they naturally contain groups that reflect real-world scenarios, such as data from the different hospitals, web sources, locations, or demographics. These natural splits make them particularly relevant for studying group distributionally robust optimization [11, 66]. Additionally, we have conducted experiments on the Cifar10 dataset as mentioned earlier.

---

> > ### Comment · Reviewer_rzqA · 2024-08-13
> >
> > I want to thank the authors for their response, which effectively addressed my concerns. Although I still have some reservations about the performance of common baselines like FedProx and SCAFFOLD in the rebuttal, I encourage the authors to provide more context on the background, clearly outline the differences between the proposed methods and the baselines, and include the new results. Given this, I’d like to raise my score to 5.

---

> > > ### Author Response · Authors · 2024-08-13
> > > **Thank you for raising the score**
> > >
> > > We are glad to hear that your concerns have been addressed by our response. As suggested, we will include more discussion on the background, differences between methods and new results. Thank you!

---

### Decision · Program_Chairs · 2024-09-25

**Decision:**

Accept (poster)

**Comment:**

Authors propose algorithms for federated distributionally robust optimization (DRO). DRO is claimed to be particularly relevant in federated setting because of the potential for non-IID data on client. They argue that existing methods have higher communication and sample complexity and provide 3 algorithms to address CVaR and KL regularized DRO frameworks, and analyze their communication complexities. They show that the algorithms have different theoretical goals and strengths. Finally authors provide multiple experimental results showing the robustness of their algorithms when compared to other federated baselines. Reviewers raised multiple concerns including lack of experimental details, lack of baselines and comparison to past literature, confusion about experimental details, unclear delineation of technical and algorithmic novelty, lack of statistical uncertainty in experiment, and too strong assumptions. Authors were able to address most of these concerns during the rebuttal phase. AC notes that paper lacks any comparison to non federated (centralized) learning methods. This is particularly important to understand the performance gap between centralized and federated setting. AC highly encourage the authors to add this baseline in the next revision.